

# Impact of Solar Geoengineering on Wildfires in the 21st Century in CESM2/WACCM6

Wenfu Tang[1], Simone Tilmes[1], David M. Lawrence[2], Fang Li[3], Cenlin He[4], Louisa K. Emmons[1], Rebecca R. Buchholz[1], Lili Xia[5]

[1]Atmospheric Chemistry Observations & Modeling Laboratory, National Center for Atmospheric Research, Boulder, CO, USA
[2]Climate and Global Dynamics Laboratory, National Center for Atmospheric Research, Boulder, CO, USA
[3]International Center for Climate and Environment Sciences, Institute of Atmospheric Physics, Chinese Academy of Sciences, Beijing, China
[4]Research Applications Laboratory, National Center for Atmospheric Research, Boulder, CO, USA
[5]Department of Environmental Sciences, Rutgers University, New Brunswick, NJ, USA

Correspondence: Wenfu Tang (wenfut@ucar.edu)

## Abstract

We quantify future changes of wildfire burned area and carbon emissions in the 21st century under four Shared Socioeconomic Pathways (SSPs) scenarios and two SSP5-8.5-based solar geoengineering scenarios with a target surface temperature defined by SSP2-4.5: solar irradiance reduction (G6solar) and stratospheric sulfate aerosol injections (G6sulfur), and explore the mechanisms that drive solar geoengineering impacts on fires. This study is based on fully coupled climate-chemistry simulations with simulated occurrence of fires (area burnt and carbon emissions) using the Whole Atmosphere Community Climate Model Version 6 (WACCM6) as the atmospheric component of the Community Earth System Model Version 2 (CESM2). Globally, total wildfire burned area is projected to increase over the 21st century under scenarios without geoengineering and decrease under the two geoengineering scenarios. By the end of the century, the two geoengineering scenarios have lower burned area and fire carbon emissions than not only their base-climate scenario SSP5-8.5 but also the targeted-climate scenario SSP2-4.5.

Geoengineering reduces wildfire occurrence through decreasing surface temperature and wind speed and increasing relative humidity and soil water, with the exception of boreal regions where geoengineering increases the occurrence of wildfires due to a decrease in relative humidity and soil water compared to present day. This leads to a global reduction in burned area and fire carbon emissions by the end of the century. However, geoengineering also yields reductions in precipitation compared to a warming climate, which offsets some of the fire reduction. Overall, the impacts of the different driving factors are larger on burned area than fire carbon emissions. In general, the stratospheric sulfate aerosol approach has a stronger fire-reducing effect than the solar irradiance reduction approach.





## 1. Introduction

Fire is an important component of the Earth system. It directly impacts climate in two main ways. First, the burning of biomass is one of the major sources of radiatively and/or chemically active trace gases and aerosols in the atmosphere (Andreae and Merlet, 2001; Li et al. 2022). Second, fires pose alterations to terrestrial ecosystem states and functioning such as changing vegetation distribution and structure, disturbing the carbon cycle and water cycle, and changing surface albedo (Bowman et al., 2009; Li and Lawrence, 2017; Liu et al., 2019; Lasslop et al. 2020). In addition to the impact on climate, fires also have significant impacts on air quality and weather across spatial scales (e.g., Bowman et al., 2009, Tang et al., 2022). For example, fires degrade air quality and human health as many of the emitted gases and aerosols from fires are primary pollutants or precursors to secondary chemically-produced pollutants (Wiedinmyer et al., 2006; van der Werf et al., 2006). Fires also alter regional dynamics and weather through changing surface heat and water vapor fluxes, convection, clouds, and precipitation (e.g., Bowman et al., 2009; Coen et al., 2013, Zhang et al., 2022).

Fire is regulated by various factors, including weather and climate conditions (e.g., soil moisture, temperature, precipitation, and wind speed), vegetation composition and structure, and human activities (e.g., land use and land cover change, human ignition and suppression) (e.g., Li et al., 2013; Chen et al., 2017; Knorr et al., 2016a, 2016b; Li et al., 2018; Pechony and Shindell, 2010; van der Werf et al., 2008). These factors also interact with each other in the Earth system (e.g., Walker et al., 2020; Loehman, 2020). For example, climate can alter vegetation composition and structure, and vegetation can also impact climate and weather through evapotranspiration. Due to the complex interactions and feedbacks among these factors and fires, quantifying and projecting the trend of fires is challenging and is subject to large uncertainties. Despite challenges and uncertainties, previous studies have generally suggested that in the future global fire risk will increase, though with significant regional differences (e.g., Abatzoglou et al., 2019; Bowman et al., 2020; Di Virgilio et al., 2019; Flannigan et al., 2009, 2013; Ford et al., 2018; Huang et al., 2015; Li et al., 2020; Liu et al. 2010; Luo et al., 2013; Pechony and Shindell, 2010; Veira et al., 2016). The growing importance combined with large uncertainties of fires has posed an urge to understand and quantify future fire trends in the context of climate change. It has been suggested that future climate mitigation should consider the impact of fires (Shiogama et al., 2020; Ward et al., 2012).

The Shared Socioeconomic Pathways (SSPs) were established to facilitate the integrated analysis of future climate impacts, vulnerabilities, adaptation, and mitigation (Riahi et al., 2017). These SSP scenarios utilized in Phase 6 of the Coupled Model Intercomparison Project (CMIP6) were generated with integrated assessment models, based on five narratives describing alternative socio-economic developments, including sustainable development (SSP1), middle-of-the-road development (SSP2), regional rivalry (SSP3), inequality (SSP4), and fossil-fueled development (SSP5). Different scenarios have different energy, land use, and emissions implications. Corresponding global population projections consistent with each of the SSPs have also been established (Jones and O'Neill, 2016).

Solar geoengineering, also known as solar radiation modification (SRM) or more generally as climate intervention, has been researched as a potential option to offset some of the radiative effects of increasing anthropogenic greenhouse gases in the future through solar radiation



modification (e.g., Kravitz et al., 2015; Tilmes et al., 2009, 2020). One proposed approach is to inject the precursor of sulfate aerosols (sulfur dioxide; $SO_2$) to the stratosphere that can reflect incoming solar radiation. To understand the impacts of sulfate aerosols compared to direct solar irradiance reduction, both experiments have been performed in parallel (e.g., Xia et al, 2016, Visioni et al., 2021a). Previous studies have analyzed the impact of geoengineering on climate outcomes (e.g., Tilmes et al., 2013, 2020; Visioni et al., 2021a). While global surface temperature targets could be reached, SRM approaches tend to overcompensate the hydrological cycle, with potential consequences to other impacts on climate and the Earth system (Robock, 2020). Since fire is a key component of the Earth system and the drivers of fires are directly or indirectly changed by solar geoengineering, the impacts of solar geoengineering on fires should also be considered when designing and assessing solar geoengineering approaches.

In this paper, we use a fully coupled Earth system model CESM2 with WACCM6 as the atmospheric component. CESM2 (WACCM6) is coupled to the Community Land Model (CLM) that includes a prognostic fire scheme, which interacts with various land and atmospheric processes. WACCM6 is currently not using biomass burning emissions derived from the land model. However, while this feedback is missing, the fire model still responds to changes in the land and atmosphere and is therefore suited to investigate how fires change in the 21st century. We analyze the future trends of burned area and fire carbon emissions under the two geoengineering scenarios and SSP scenarios, and then analyze how the two solar geoengineering approaches impact fire activity. This paper is organized as follows: Section 2 describes the model simulations; Section 3 presents the future trends of burned area and fire carbon emissions under SSP scenarios and geoengineering scenarios. Section 4 discusses how geoengineering impacts fire, and Section 5 concludes the study.

## 2. Model descriptions and simulations

### 2.1 CESM2 (WACCM6)

CESM2 (WACCM6) is a community model that has components of ocean, atmosphere, land, sea-ice, land-ice, river, and wave models. These components are coupled in CESM2 by exchanging states and fluxes via a coupler (Danabasoglu et al., 2019). The Community Land Model Version 5 (CLM5) is the land component of CESM2 (Lawrence et al., 2019). CLM uses prescribed temporal land use and land cover change (LULCC), which consists of an annual time series of the spatial distribution of the naturally vegetated and cropland units of each grid cell, combined with the distribution of plant functional types (PFTs) and crop functional types (CFTs) existing in those land units (Lawrence et al., 2019). The interactive fire scheme in the CLM5 is a key component of this study and is described in more detail in Section 2.2. WACCM6 is a high-top atmospheric model with 70 vertical levels and model top at ~140 km, therefore it has reasonable representation of the stratosphere. WACCM6 also includes comprehensive chemistry and aerosol mechanisms (Gettelman et al., 2019; Emmons et al., 2020, Tilmes et al., 2019).

### 2.2 Description and evaluation of fire scheme in CESM2/CLM5



126  The fire scheme in CESM2/CLM5 accounts for four types of fires: agricultural fires in
127 cropland, deforestation fires in the tropical closed forests, peat fires, and non-peat fires outside
128 cropland and tropical closed forests (Li et al., 2012, 2013). Agricultural fire is accounted for in
129 these simulations but is not included in the analysis, since we focus on wildfires here. In the fire
130 scheme, burned area is affected by climate and weather conditions, vegetation composition and
131 structure, and human activities. Climate and weather conditions (e.g., temperature, precipitation,
132 wind, humidity, and soil moisture) impact natural and human ignition and fire spread through fuel
133 availability and fuel combustibility. Human activities impact deforestation fires via deforestation
134 rates that are applied from the Land Use Harmonization dataset (LUH2, Hurtt et al., 2020) that is
135 used in these experiments. Human impacts on non-deforestation and non-peat fires include both
136 ignition and suppression and are parameterized as functions of both population density and Gross
137 Domestic Product (GDP). In our setup, the global population scenarios corresponding to SSP
138 scenarios (Jones and O'Neill, 2016) are used while regionally-explicit GDP was held constant for
139 all WACCM6 simulations analyzed in this study. Fire-induced changes (including biomass and
140 peat burning, vegetation mortality, adjustment of the carbon and nitrogen (C/N) pools, carbon
141 emissions, changes in vegetation structure and functioning as well as surface water and energy
142 fluxes) are then simulated based on the calculated burned area (Li et al., 2012, 2013). These fire-
143 induced surface property changes in the land model further alter atmospheric states (i.e.,
144 temperature and water vapor) in the coupled model. Although the burned area and fire carbon
145 emissions are simulated in CLM5, our CESM2/(WACCM6) simulations use prescribed fire
146 emissions based on the CMIP6 projected inventories for trace gases and aerosols (Riahi et al., 2017)
147 for different SSPs and geoengineering scenarios. Full coupling of simulated fire aerosol emissions
148 is an area of ongoing development and analysis with the CESM project.

149  The fire scheme in CESM has been validated and evaluated in both uncoupled and coupled
150 versions (Li et al., 2012, 2013, 2017, 2018; Li and Lawrence 2017) and compared with other fire
151 models within the Fire Modeling Intercomparison Project (FireMIP) (Li et al., 2019). Evaluation
152 results have shown that the fire scheme can reasonably reproduce the observed amount, spatial
153 pattern, seasonality, and interannual variability of global fires, and fire-population relationship
154 under present-day climate, and has a similar historical long-term trend to the multi-source merged
155 historical reconstructions used as input data for CMIP6 (Li et al. 2018, Li et al. 2019). Although
156 the model underestimates the climate impacts on fires in boreal North America, it still performs
157 better than many other fire models (Yue et al., 2016). Here we briefly evaluate the fire carbon
158 emissions from the CESM2 (WACCM6) simulations with two satellite-based fire emission
159 inventories, namely FINNv2.5 (Fire INventory from NCAR Version 2.5; Wiedinmyer et al., 2022)
160 and GFED4.1s (Global Fire Emissions Database, Version 4.1s; Randerson et al., 2018). The annual
161 total emissions and global distributions of WACCM simulations agree well with those from
162 FINNv2.5 and GFED4.1s (Figures S1 and S2). The annual total fire carbon emissions during 2015-
163 2019 estimated from the WACCM simulations (2.5 PgC/yr) fall into the range of GFED4.1s (2.0
164 PgC/yr) and FINNv2.5 (3.8 PgC/yr).

165 **2.3 SSPs and geoengineering scenarios**

166  The Scenario Model Intercomparison Project (ScenarioMIP) based on SSPs is the primary
167 activity within CMIP6 that provides multi-model climate projections based on alternative
168 scenarios (O'Neill et al., 2016). These climate projections are driven by SSP scenarios and are
169 related to the Representative Concentration Pathways (RCPs) as described below. The Land Use



Model Intercomparison Project (LUMIP) also provides LULCC data for SSPs (Lawrence et al.,
2016, Hurtt et al., 2020). In this study, the SSP1-2.6, SSP2-4.5, SSP3-7.0, and SSP5-8.5 scenarios
(O'Neill et al., 2016) are shown. (1) SSP1-2.6 (sustainable development) is the low end of the
range of future forcing pathways in SSP and updates the RCP2.6 scenario. SSP1 includes
substantial land use change, particularly with increasing global forest cover. (2) SSP2-4.5 is a
scenario that represents the middle part of the range of future forcing pathways and updates the
RCP4.5 scenario. Land use and aerosol changes in SSP2 (middle-of-the-road development) are
not extreme relative to other SSPs. (3) SSP3-7.0 is a scenario with both substantial land use
changes (particularly decreased global forest cover) and high near-term climate forcers emissions,
particularly sulfur dioxide ($SO_2$). (4) SSP5-8.5 is the unmitigated baseline scenario, representing
the high end of the range of future pathways, and updates the RCP8.5 scenario. There is relatively
little land-use change in the 21st century in this scenario which leads to slow decline in the rate of
deforestation (O'Neill et al., 2017).
The Geoengineering MIP Phase 6 (GeoMIP6) proposed experiments for future projection with
geoengineering measures implemented based on ScenarioMIP. In this study we also analyze the
response of wildfires under two of the geoengineering experiments – G6Sulfur and G6Solar
(Kravitz et al., 2015). Both of these geoengineering scenarios aim to reduce forcing from
ScenarioMIP Tier 1 high forcing scenario (SSP5-8.5) to the medium forcing scenario (SSP2-4.5),
going from 8.5 to 4.5 $Wm^{-2}$ in 2100.
G6Sulfur reduces forcing with stratospheric sulfate aerosols. In G6Sulfur experiment, $SO_2$, the
precursor of stratospheric sulfate aerosol has been continuously injected into the model at 25 km
altitude at the Equator with the goal of reducing the magnitude of the net anthropogenic radiative
forcing and reaching surface temperatures at SSP2-4.5 levels.
G6Solar uses the same setup as G6sulfur, but uses solar irradiance reduction to reduce the
magnitude of the net anthropogenic radiative forcing. The reduction of the solar constant in
G6Solar and the injected $SO_2$ in G6Sulfur is determined by a feedback algorithm described in
Kravitz et al. (2017) and used in Tilmes et al. (2018, 2020).
**2.4 Simulations**
In this study we analyze results from fully coupled WACCM6 simulations for future projection
under the aforementioned scenarios from GeoMIP and ScenarioMIP. The continuous long-term
(2015 to 2100) simulations used in this study provide a continuous picture of future fire changes
and allow us to investigate when and how major changes in the fire trends occur. The horizontal
resolution for land and atmosphere is $1.25° \times 0.9°$ (longitude × latitude). Multiple simulations (2~5
members) are conducted for each scenario except for the SSP1-2.6 and SSP3-7.0 scenarios (see
Table S1). WACCM6 historical simulations serve as initial conditions for the future scenarios.
Future climate under these simulations has been analyzed in Meehl et al. (2020) and Jones et al.
206    (2020).

**3   Future trends of fires**
**3.1 Future trends of burned area and fire carbon emissions under the SSP scenarios**



The global total wildfire burned area in these simulations is projected to increase under all
the SSP scenarios (Figure 1a). The largest increases in the global burned area are seen in the SSP5-
8.5 scenarios (~20%) and SSP3-7.0 (~10%). The changes in the other scenarios are relatively small
(Table S2). In terms of the spatial distribution, the 40°N–70°N latitude is the only latitude band in
which the burned area consistently increases under all the SSP scenarios (Figure 1b). In the 10°S–
5°N latitude band (tropical region), the burned area consistently decreases under all scenarios to a
diverse extent. A more detailed discussion on future trends of fire activity under the SSP scenarios
are provided in the Supplement.
**3.2 Future trends of burned area and fire carbon emissions with geoengineering**
The two geoengineering scenarios (G6Sulfur and G6Solar) are based on SSP5-8.5 and
targeted SSP2-4.5. As G6Sulfur reduces the forcing through stratospheric sulfate aerosols while
G6Solar directly decreases total incoming solar irradiance, the difference between the two provides
insight on the other impacts of sulfate aerosols on fires besides the forcing change. Even though
fire carbon emissions are largely driven by burned area, they are also impacted by fuel availability
and combustion completeness. Therefore, the fire carbon emissions and burned area generally
show trends consistent with burned area, with some notable differences. Both burned area and fire
carbon emissions under the two geoengineering scenarios are lower than those under SSP5-8.5
(Figures 2a and 2c). Lower fire activity in these geoengineering scenarios than SSP5-8.5 is
expected due to reduced surface warming towards SSP2-4.5 target climate conditions. However,
we found that by the end of the century, the two geoengineering scenarios have lower burned area
and fire carbon emissions than not only their base-forcing scenario SSP5-8.5 but also the targeted-
forcing scenario SSP2-4.5 (Figures 2a and 2c). The change of the two geoengineering scenarios
compared to SSP2-4.5 by the end of the century is small in burned area (-2% – -12%) but relatively
large in fire carbon emissions (-18% – -23%). However, when compared to SSP5-8.5, the
reduction of the two geoengineering scenarios in burned area (-18% – -26%) is similar to that in
fire carbon emissions (-20% – -26%). This implies that the difference in fire carbon emissions
between the two geoengineering scenarios and SSP2-4.5 are less driven by burned area and that
fuel availability plays a more important role in this comparison, while for the difference to SSP5-
8.5, changes in burned area plays more of a role in emission differences. The two geoengineering
approaches (G6solar and G6sulfur) generally lead to reduced fire activity compared to SSP5-8.5
in most regions in 2091-2100, except for Northern Hemisphere Africa and Equatorial Asia
(Figures S3 and S4). When comparing the period 2091-2100 to the period 2021-2030, the largest
decrease in global total wildfire burned area is seen in the G6sulfur scenario among all the
scenarios in this study (~ -11%; see Table S2).
In the 40°N–70°N latitude band, the burned area consistently increases under not only all
the SSP scenarios but also the two geoengineering scenarios when comparing the period 2091-
2100 to the period 2021-2030 (Figure 2b). However, the increase in burned area is lower in the
two geoengineering scenarios compared to SSP5-8.5 and is similar to the SSP2-45 scenario. In the
-20°S–0° latitude band, the reduction in burned area is larger under G6sulfur than that under
G6Solar (Figure 2a). Generally, G6sulfur has a stronger fire-reducing effect than G6solar, with
exceptions such as over Europe. We also found notable differences between the two
geoengineering methods for some specific regions, implying that the geoengineering method
chosen could be inequitable for some countries. For example, G6Solar is the better choice for


producing less burned area in Europe, while over Southern Hemisphere Africa, G6Sulfur is better
than G6Solar (see Figure S4).

## 4 Mechanism of geoengineering impacting fires

The two SSP5-8.5-based geoengineering scenarios successfully reduce the radiative
forcing from 8.5 Wm$^{-2}$ (as in SSP5-8.5) to 4.5 Wm$^{-2}$ (as in SSP2-4.5) in 2100 and global surface
temperatures between SSP2-4.5 and the two geoengineering scenarios are nearly the same.
However, both geoengineering scenarios produce less fire than SSP2-4.5 by 2100 (Figures 2 and
3). There are different processes involved in the cooling in G6Sulfur (due to the stratospheric
sulfate aerosols) and the cooling in G6Solar (due to directly reduced insolation) (Visioni et al.,
2021a). Because of the difference in the resulting climate response, these two geoengineering
approaches impact fires differently, even though they are designed to achieve the same forcing
level by 2100. Previous studies indicate that stratospheric heating caused by aerosols can impact
precipitation and temperature at the surface through alterations to stratospheric dynamics (Jiang et
al., 2019; Simpson et al., 2019; Richter et al., 2017; Visioni et al., 2020). Last but not least, the
two geoengineering approaches also result in different outcomes for other quantities important for
fires. For example, enhanced stratospheric aerosol burden results in changes in direct to diffuse
light which promotes plant growth (e.g., Xia et al., 2017; Xu et al., 2020). On the other hand, it
can reduce in the hydrological cycle and regional precipitation changes due to the aerosol heating
effects in the lower tropical stratosphere (e.g., Tilmes et al., 2013, Simpson et al., 2019).
Here we analyze the key variables in the Earth system that are involved in the processes
from the reduced insolation on the top of the atmosphere and sulfate aerosols in the stratosphere
to fires at the surface. Note that hereafter for a scenario with multiple ensemble members, only the
ensemble mean is analyzed and shown. The key variables shown in this section are selected via
comparing the key variables that determine fire activity in the fire scheme in CESM2/CLM5 with
the key climate variables that are impacted by geoengineering approaches. The analyses are
conducted for 14 individual fire regions following Giglio et al. (2010), namely Boreal North
America, Temperate North America, Central America, Northern Hemisphere South America,
Southern Hemisphere South America, Europe, Middle East, Northern Hemisphere Africa,
Southern Hemisphere Africa, Boreal Asia, Central Asia, Southeast Asia, Equatorial Asia, and
Australia and New Zealand (Figure S3).

## 4.1 Surface temperature

Even though the mean surface temperature (TS) for the whole globe and the land are similar
under the two geoengineering scenarios and SSP2-4.5 (Figure 4), regional differences exist
(Figures 5). For example, over Equatorial Asia, the annual surface mean temperatures in the two
geoengineering scenarios are consistently lower than that in SSP2-4.5 by ~0.3K during 2091-2100
(Figure S6). The spatial distribution of burned area difference and fire carbon emission difference
between G6Solar/G6Sulfur and SSP5-8.5 (Figure 3) are not always co-located with their spatial
distribution of surface temperature difference (Figure 5). To understand to what extent the surface
temperature drives fire activity change, we calculate correlations of surface temperature change
and burned area/fire carbon emission change for individual fire regions under SSP2-4.5, G6Solar,
and G6Sulfur. Surface temperature change (ΔTS) for a given region is calculated based on the
individual model grids within the region and annual values between 2091-2100. It is defined as



the difference between the analyzed scenario (i.e., G6Solar, G6Sulfur, and SSP2-4.5) and the
reference scenario (i.e., SSP5-8.5). Burned area change (ΔBA) and fire carbon emission change
(ΔCemis) are defined in the same way. The correlations calculated here account for spatial
variability within the region and interannual variability during 2091-2100.
Overall, surface temperature plays a more important role in the decrease of fire activity in
the two geoengineering scenarios compared to that in SSP2-4.5 relative to SSP5-8.5 (Figure 6).
This is expected because the only difference between the two geoengineering scenarios and SSP5-
8.5 is the specific application of climate intervention; whereas the differences between SSP2-4.5
and SSP5-8.5 involves several other differences including population growth and LULCC. For
G6Solar and G6Sulfur, the strongest impact of surface temperature change on burned area occurs
over Southern Hemisphere South America (correlation=0.42 for G6Solar and 0.45 for G6Sulfur),
followed by Southern Hemisphere Africa, Temperate North America, and Europe. The impact of
surface temperature change over boreal regions (Boreal North America and Boreal Asia) are
relatively small. This suggests that the changes in area burnt in these regions are not predominantly
driven by the surface temperature changes, but by other factors. For G6Solar and G6Sulfur, the
impact of surface temperature on burned area is generally larger than its impact on fire carbon
emissions. This is expected as fire carbon emissions in CESM2/WACCM6 are determined by
burned area together with vegetation characteristics (carbon density and combustion completeness;
Li et al., 2012), which introduces more uncertainties. The only exception occurs over the Northern
Hemisphere South America where surface temperature plays a more important role in fire carbon
emissions than burned area for not only G6Solar (correlation is 0.37 versus 0.29) and G6Sulfur
(correlation is 0.37 versus 0.24) but also SSP2-4.5 (correlation is 0.40 versus 0.23). Over Northern
Hemisphere South America, the correlations between ΔTS and ΔBA/ΔCemis are also close under
the three scenarios. Since combustion completeness is a fixed parameter, this difference points to
the possibility that the reduced surface temperature has a larger impact on carbon density over
Northern Hemisphere South America than over other regions.
Overall, we find that the surface temperature change introduced by the two geoengineering
approaches (solar irradiance reduction and stratospheric sulfate aerosols) by the end of the century
impacts burned area and fire carbon emissions, e.g., the introduced cooling results in smaller fire
activity. The degree of impact varies dramatically across different regions. The impact of surface
temperature in G6Solar and G6Sulfur are overall close. However, surface temperature alone does
not account for all the changes in fire activity.
**4.2 Precipitation**
Precipitation change is also an important consequence of climate change and
geoengineering (Figure 4). Global precipitation is expected to increase under climate change as
higher tropospheric temperature leads to more moisture in the air. Previous studies found that
geoengineering could eliminate these increases in precipitation and can even reduce global mean
or regional precipitation relative to the target scenario, depending on the geoengineering approach
(Tilmes et al., 2013, Simpson et al., 2019, Visioni et al., 2021a). The spatial distribution of
precipitation changes under G6Solar and G6Sulfur relative to SSP5-8.5 are similar (Figure 5). The
trend of precipitation varies dramatically across regions (Figure S8). Precipitation is also important
for fires. Precipitation itself could have either a positive or a negative impact on future fires
because precipitation can impact both fuel combustibility and fuel availability, which impact fire


in opposite directions. In addition, precipitation changes can also lead to changes in relative
humidity and soil water content, which are important factors for fires. Here we apply the same
analyses for precipitation change (ΔPrecip) as in Section 4.1 for surface temperature change (ΔTS).
The reduction in precipitation by geoengineering has the opposite impact on fire as the
reduction in surface temperature by geoengineering, as shown by the negative correlations of
ΔPrecip and ΔBA/ΔCemis (Figure 6). The correlations are consistently negative across all the
scenarios (G6Solar, G6Sulfur, and SSP2-4.5) and almost all regions. The largest impact of
precipitation change occurs over Equatorial Asia for all three scenarios (correlation is -0.45–-0.42
for ΔBA and -0.43–-0.33 for ΔCemis), which is aligned with the strong precipitation change over
the region (Figures 5). Over the Middle East, precipitation change has a relatively large impact on
burned area and fire carbon emissions under G6Solar as well as SSP2-4.5, however the impact is
small under G6Sulfur. We note that unlike the impact of ΔTS, the impact ΔPrecip is relatively
large over boreal regions. We conduct a sensitivity test of 1-year lag correlation to understand the
impact of previous year precipitation change on fire activity (for example calculating correlation
of ΔPrecip for 2091 and ΔBA/ΔCemis for 2092). We found that this correlation is still significant
for most regions, though it is generally lower. Overall precipitation change is inversely related to
burned area change and fire carbon emission change. Therefore, for these regions where
precipitation is reduced compared to SSP5-8.5 as a consequence of geoengineering such as
Equatorial Asia, the reduction in burned area and fire carbon emissions due to reduced surface
temperature are offset to some extent.
**4.3 Humidity**
Humidity is also impacted by geoengineering. The future trends of specific humidity (g/kg)
and relative humidity (%) are opposite as specific humidity is projected to increase while relative
humidity is projected to decrease compared to SSP5-8.5 (Figure 4). Their spatial distribution and
inter-scenario differences are also divergent (Figures 4 and 5). This is due to the fact that relative
humidity is driven by not only the actual moisture content but also the temperature. The same
amount of water vapor results in a higher relative humidity in colder air than in warm air. Therefore
a reduction in relative humidity in a warming climate indicates that the relative amount of water
vapor has not increased proportional to the warming. Relative humidity is a driving variable in the
CLM5 fire module in multiple places (e.g., lower relative humidity leads to higher fuel
combustibility and larger fire spread). Here we focus our analysis on the relative humidity change
at 2-meter (ΔRH) as relative humidity is directly used in the CLM5 fire module. Changes in
relative humidity show different spatial distribution between the G6solar minus SSP5-8.5 and
G6sulfur minus SSP5-8.5 (Figure 5), even though their global average values are close (Figure 4).
Since 2-meter relative humidity is strongly driven by evapotranspiration, the difference between
G6sulfur and G6Solar points to the possibility that stratospheric sulfate aerosols lead to more
scattered light and hence enhanced plant growth than the solar case, which results in more
evapotranspiration.
The relative humidity change (ΔRH) is negatively correlated to ΔBA/ΔCemis across all
scenarios and regions (Figure 6). Therefore, the higher relative humidity in G6Solar, G6Sulfur,
and SSP2-4.5 than SSP5-8.5 (Figure 4) leads to less fire activity globally. Overall, the relative
humidity change is more strongly correlated to ΔBA/ΔCemis, indicating that relative humidity





change is a more important driver of fire activity change under geoengineering than surface
temperature or precipitation.
**4.4 Wind speed**
Wind speed is also an important driving factor in fire spread and is also indirectly impacted
by geoengineering (Figure 4). In CLM5, wind speed is used in the calculation of fire spread and
hence burned area. Wind speed mainly has an indirect impact on fire carbon emissions through
burned area. Here we analyze 10-meter wind speed (U10). By the end of the century, SSP2-4.5
has slightly higher U10 than SSP5-8.5, G6Solar has similar U10 as SSP5-8.5, while G6Sulfur has
slightly lower U10 than SSP5-8.5 over land (Figure 4). However, the regional difference can be
relatively large (Figures 5). G6sulfur and G6solar have significantly different U10 over Southern
Hemisphere ocean (Figures 5). However, the difference in U10 between G6solar and G6sulfur
over land is relatively small with exceptions such as over Australia and Northern Hemisphere
Africa where G6sulfur has lower U10.
Wind speed change has consistently positive correlations with changes in burned area and
fire carbon emissions under the two geoengineering scenarios across all analyzed regions (which
is not the case for SSP2-4.5, where $\Delta$U10 is negatively correlated $\Delta$BA or $\Delta$Cemis over most
regions). This indicates that the reduction in wind speed as a byproduct of geoengineering (Figure
4) leads to less fire activity globally. The wind speed reduction is relatively large over South
Hemisphere Africa (Figure 5), and the correlations are also high, indicating the wind speed
reduction is partially responsible for the reduction in fire activity over South Hemisphere Africa.
**4.5 Soil water content**
Soil water content is a key driver of fire activity as it impacts fuel combustibility and fire
spread. Soil water content is indirectly impacted by the geoengineering approaches through the
hydrological cycle. The precipitation changes as a result of geoengineering compared to SSP5-8.5
strongly impacts the soil water content, and the soil water content further drives the relative
humidity near the surface through evapotranspiration. We see a much smaller reduction in soil
water content in the geoengineering runs compared to SSP2-45. Therefore, the future trends of soil
water content (here we use the model variable SOILWATER_10CM, i.e., the soil water content in
the top 10 cm (kg/m$^2$) to evaluate soil moisture) are close to the future trends of relative humidity
(Figure 4) globally. However, in the last decade of the century, difference in soil water content
among the scenarios is larger than the difference in relative humidity among the scenarios (the
difference of the 3 scenarios from SSP5-8.5 are ~1–2% for relative humidity and ~4%–7% for
SOILWATER_10CM). Here we include analyses of soil water content not only because it is a
very important driver of fire activity but also because the spatial distributions of soil water change
($\Delta$SOILWATER) can be different than relative humidity change in some regions (Figures 5).
Overall, similar to precipitation and relative humidity, soil water content change is negatively
related to burned area and fire carbon emissions with different spatial distributions (Figure 6). For
example, over the boreal regions and Europe, the impact of $\Delta$SOILWATER is smaller than the
impact of $\Delta$RH, while over Central Asia it is larger.
**4.6 Others**





There are other relevant variables that are not analyze in detail here. For example, the
reduction in the downwelling solar flux at the surface (ΔFSDS) is a direct consequence of
geoengineering (solar irradiance reduction and stratospheric sulfate aerosols). In addition, water
vapor content and cloud change as a consequence of geoengineering also impact downwelling
solar flux at the surface. We include the analyses of downwelling solar flux in the supplement
(Figures S9-S10) as the downwelling solar flux at the surface does not directly determine burned
area and fire carbon emissions in the model. The downwelling solar flux at the surface is positively
related to burned area and fire carbon emissions. Therefore, the lower downwelling solar flux at
the surface than SSP5-8.5 as a result of the geoengineering approaches leads to less fires globally
while the higher downwelling solar flux at the surface under SSP2-4.5 than SSP5-8.5 tends to
increase fire activity and can offset the overall reduction fires in SSP2-4.5 than SSP5-8.5 to some
degree. As another example, vegetation carbon can also impact the total fire carbon emissions and
are also impacted by fire activity. However, we do not further analyze the impact of fuel load
because geoengineering approaches do not seem to change global total fuel load significantly. The
future trend of total vegetation carbon under G6Solar and G6Sulfur are very close to SSP5-8.5,
and the three of them are different from SSP2-4.5 as total vegetation carbon is largely driven by
$CO_2$ (Figure 4).

**4.7 G6Sulfur versus G6Solar**

Comparisons between G6Sulfur and G6Solar provide insight on the potential impact of
stratospheric sulfate aerosols on fires other than the intended climate intervention. In general, using
sulfur to create climate control enhances the effect of the solar management on the modeled fire
response. While both geoengineering approaches show strongest inverse relationships between fire
parameters and relative humidity and soil moisture, G6Sulfur shows smaller reductions in these
climate variables than G6Solar. Globally, G6Sulfur has lower burned area and fire carbon
emissions than G6Solar by the end of the century. The differences between G6Sulfur and G6Solar
varies regionally (Figures 7a-7b). For example, over most regions, G6Sulfur has less fire activity
than G6Solar whereas over Europe, G6Sulfur has more fire activity than G6Solar, which is related
to the warming over Northern Eurasia caused by G6Sulfur (Figure 7c) and a positive correlation
between BA and surface temperature over Europe. However, we note that two ensemble members
may not fully reflect the robust signal. The spatial distributions of differences between G6Sulfur
and G6Solar in burned area and fire carbon emissions (Figures 7a-7b) are close to the spatial
distributions of difference between G6Sulfur and G6Solar in relative humidity (Figure 7e) and soil
water content (Figure 7g). G6Sulfur has higher relative humidity and soil water content over most
regions. However, over Europe relative humidity and soil water content in G6Sulfur are lower than
those in G6Solar, which is consistent with what has been found in burned area and fire carbon
emissions. In addition, over South America, the distribution of difference in relative humidity and
soil water content is similar to the distribution of difference in burned area and fire carbon
emissions. This indicate that the differences in future fire activity between the two geoengineering
approaches is likely driven by relative humidity and soil water content.
A summary of the relationships between ΔBA and the changes in the related variables (ΔTS,
ΔPrecip, ΔRH, ΔU10, ΔSOILWATER, and ΔFSDS) for G6Sulfur versus G6Solar is shown in
Figure 8 (note that ΔBA as well as Δ of other variables are calculated by the difference of the
geoengineering run from the reference case, i.e., SSP5-8.5). Overall, the impacts of these driving
variables are similar in the two geoengineering approaches (as the points fall close to the diagonal).



However, these variables in general have larger impacts on burned area in G6Solar than in
G6Sulfur (as the majority of the points fall in the shaded area where the x-axis value is larger than
the y-axis value). This is expected since the climate impacts of solar irradiance reduction (G6Solar)
is more direct than that of stratospheric sulfate aerosols (G6Sulfur) and stratospheric sulfate
aerosols can yield to additional changes (such as higher diffuse radiation that benefits plant
growth). This is consistent with that G6Sulfur has slightly higher total vegetation carbon than
G6Solar or SSP5-8.5, even though this difference is relatively small compared to the difference
caused by $CO_2$ (Figure 4g).
**4.8 Discussion**

472        The key finding of this study is that fire burned area and emissions are lower in the
geoengineering runs than not only SSP5-8.5 but also the target SSP2-4.5 run in CESM2/WACCM6.
Here we analyze the key climate variables that are largely and/or directly impacted by the two
geoengineering approaches and are important drivers of fires. A summary of the relationships
between ΔBA and the change in the related variables (ΔTS, ΔPrecip, ΔRH, ΔU10, ΔSOILWATER,
and ΔFSDS) versus the relationships between ΔCemis and the change in the related variables for
G6Solar, G6Sulfur, and SSP2-4.5 are shown in Figure 9. The future trends of the analyzed
variables and their changes from SSP5-8.5 can be opposite over different regions. However, the
directions of impact (i.e., positive or negative correlation) are overall consistent across the 14 fire
regions and 3 scenarios. Therefore the dominant factors are also different across regions.

482        We note that under both geoengineering scenarios, changes in relative humidity, soil water,
and downwelling solar flux at the surface all have strongest impacts over Equatorial Asia (as
shown by strongest correlations among the 14 regions; Figure 9). Changes in wind speed and
precipitation also have relative strong impacts over Equatorial Asia compared to other regions.
Overall, Equatorial Asia is the most sensitive to the climate variable changes introduced by both
geoengineering approaches (Figure 9), even though the resulting fire activity changes over
Equatorial Asia are not as strong as some other regions (Figure 3) likely due to the relatively weak
change in the climate variables (e.g., Figures 5). On the contrary, Boreal North America is not
sensitive to most of the climate variable changes introduced by both geoengineering approaches
(the correlations are the lowest and close to 0, Figure 9), which is likely the reason why the 40°N–
70°N latitude band is the only latitude band in which the zonal mean burned area consistently
increases even under the geoengineering scenarios (Figures 1 and 2). Boreal Asia is similar to
Boreal North America with the correlations overall being slightly stronger.

495        For G6Solar and G6Sulfur, the impacts of the shown variables (especially for ΔTS, ΔRH,
ΔU10, and ΔFSDS) on burned area are in general stronger than their impacts on fire carbon
emissions (as shown by more data points that fall into the shaded area). This is expected because
these variables first impact burned area, and then fire carbon emissions are determined by burned
area and fuel availability. Fuel availability is further directly or indirectly impacted by many other
variables including the shown ones here, which introduce more uncertainties. The patterns in
G6Solar and G6Sulfur and closer to each other when using SSP2-4.5 as a reference (Figures 6).
This is not only because their approaches to reducing forcing from SSP5-8.5 to 4.5 $W/m^2$ are
different, but also because the scenario configuration of SSP2-4.5 is different from SSP5-8.5 and
SSP5-8.5-based G6Solar and G6Sulfur (e.g., LULCC).





505       The analyses above (Sections 4.1-4.7) use SSP5-8.5 as the reference case to calculate the
changes (Δ) because the two geoengineering scenarios are based on SSP5-8.5, and their difference
is only due to the geoengineering approaches. Here we also include analyses that uses the target
SSP2-4.5 as the reference case in the Supplement (Figures S13). The signs of the correlations are
in general consistent whether SSP5-8.5 or SSP2-4.5 is used as the reference case (Figures S14-
S15). For example, even though relative humidity change from SSP2-4.5 are very different
regionally under G6Solar and G6Sulfur (Figure 5), the signs of the correlations are consistently
negative over all regions and under the two geoengineering scenarios. In general, the impacts of
the analyzed variables on changes of the burned area and fire carbon emissions from SSP2-4.5 are
weaker (Figures S14-S15), likely due to the fact that the changes (Δ) between the two
geoengineering scenarios and SSP2-4.5 are due to not only geoengineering introduced climate
variable changes (e.g., surface temperature, relative humidity, soil water content, etc.) but also
other factors such as atmospheric $CO_2$ and LULCC.
**4.9 Uncertainty and limitation**

519       We recognize that there are several limitations in this study. For example, even though
CESM2 is a state-of-the-art model, uncertainties and limitations exist in the model
parameterizations (including the parameterization of fire-related processes and the lack of
interactive fire emissions). In addition, the fire emissions of trace gases and aerosols are not fully
coupled, as CESM2 uses the CMIP6 fire emission inventories. This study analyzes results from
only one model (CESM2) and similar studies need to be conducted with other models to test inter-
model consistency. Lastly, there are only two ensemble members in each geoengineering scenario,
which can lead to larger variability at regional scale in particular resulting in large uncertainties in
the response of geoengineering on rainfall with implications of other relevant variables. While
largescale changes are significant, a larger ensemble size in future study will reduce uncertainties
in the regional results. More studies are needed to fully understand the future trends of fires and
the impact of geoengineering on fires.
**4. Conclusions**

532       Here we analyzed the future fires under geoengineering as well as SSP scenarios, and
assess how the different geoengineering approaches impact fires. The major conclusions and
implications are as follows:
(1) The global total wildfire burned area is projected to increase under the unmitigated scenario
(SSP5-8.5), and decrease under the two geoengineering scenarios (solar irradiance reduction and
stratospheric sulfate aerosols) in the 21st century.
(2) By the end of the century, the two geoengineering scenarios exhibit lower burned area and fire
carbon emissions than not only their base-forcing scenario (SSP5-8.5) but also the targeted-forcing
scenario (SSP2-4.5).
(3) The two geoengineering approaches (solar irradiance reduction and stratospheric sulfate
aerosols) generally lead to less wildfire activity in most regions in 2091-2100, except for the
Northern Hemisphere Africa and Equatorial Asia. The 40°N–70°N latitude band is the only
latitude band in which the zonal mean burned area consistently increases under all the scenarios,
even the geoengineering scenarios.



(4) Overall, changes of G6Solar and G6Sulfur from SSP5-8.5 in surface temperature, wind speed,
and downwelling solar flux at the surface are positively correlated to the changes in burned area
and fire carbon emissions, while their changes in precipitation, relative humidity, and soil water
content are negatively correlated to the changes in burned area and fire carbon emissions.
(5) Generally, the stratospheric sulfate aerosols approach has a stronger fire-reducing effect than
the solar irradiance reduction approach. The impacts of the analyzed variable changes are generally
larger (percent-wise) on burned area than fire carbon emissions.
(6) Geoengineering imposed reduction in surface temperature and wind speed, and increase in
relative humidity and soil moisture, reduce fires by the end of the century. However, the reduction
in precipitation resulting from geoengineering offsets its overall fire-reducing effect to some extent.
The success of future fire mitigation with the two geoengineering approaches in the
CESM2/WACCM6 model results is encouraging. However, this study is not a closure study due
to the uncertainties and limitations (Section 4.9). More research is needed for this topic. Here we
do not indicate that fewer fires under the geoengineering approaches are definitively beneficial.
After all, fire is a natural process and a key component of the dynamic Earth system, and wildfires
were present long before anthropogenic activities. Lastly, fire risk increase is only one of many
possible consequences of climate change, and fire activity reduction is also only one of many
possible consequences of climate intervention. We present this study only as a reference for the
future when geoengineering is considered.

**Data availability**
The simulation data used in this study are archived on the Earth System Grid Federation (ESGF)
(https://esgf-node.llnl.gov/projects/cmip6; last access: 12 December 2022). The model Source ID
is CESM2-WACCM for CESM2-WACCM6. FINN2.5 data are available at:
https://www.acom.ucar.edu/Data/fire/. GFED data are available at:
https://www.globalfiredata.org/.

**Author contributions**
WT led the analysis with the contribution from ST. ST and DML contributed to the interpretation
of the model results. WT prepared the paper with improvements from ST, DML, FL, CH, LKE,
RRB, and LX.

**Acknowledgements**
This material is based upon work supported by the National Center for Atmospheric Research,
which is a major facility sponsored by the National Science Foundation under Cooperative
Agreement No. 1852977. Wenfu Tang was supported by NCAR Advanced Study Program
Postdoctoral Fellowship. W. Tang thanks Wangcai Bao (Syrian hamster; Sep 8, 2020 – Jul 22,
2022) for his support during the pandemic.

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

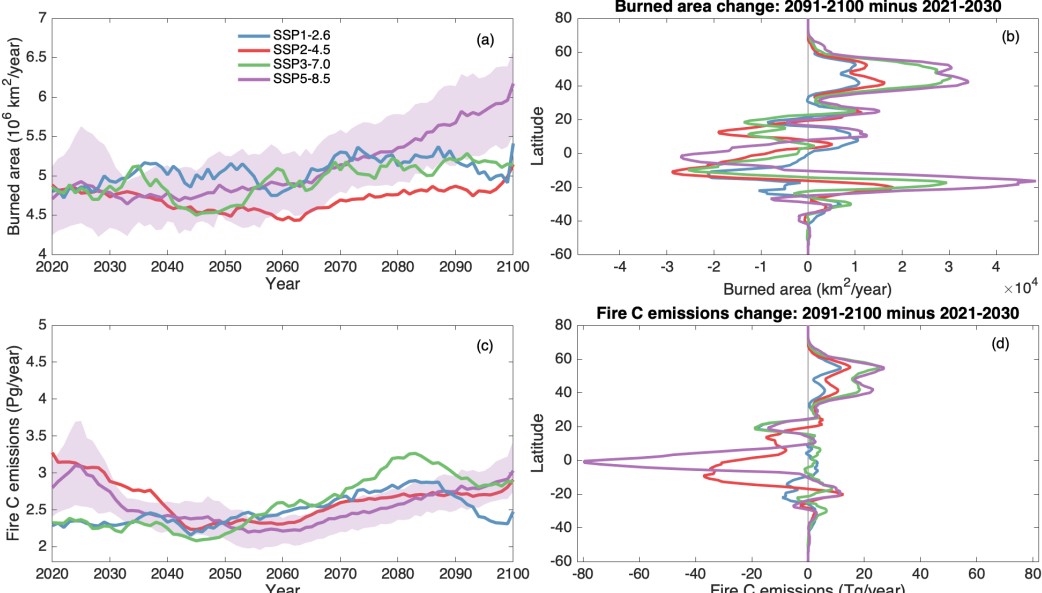

**Figure 1.** Overall global burned area and fire carbon emission trends and changes under SSP
scenarios. (a) Time series of global burned area from 2020 to 2100 under the SSP1-2.6, SSP2-4.5,
SSP3-7.0, and SSP5-8.5 scenarios (represented by different colors). For the scenarios with
multiple simulations, the ranges are also shown by the shaded areas. The time series are shown as
5-year moving averages. (b) Zonal changes (absolute value) of burned area in the period 2091-
2100 relative to the period 2021-2030 (calculated by the value in 2091-2100 minus the value in
2021-2030), under the SSP1-2.6, SSP2-4.5, SSP3-7.0, and SSP5-8.5 scenarios (represented by
different colors, color code is the same as it in panel a). 5-degree moving average were applied to
the shown zonal changes. Panels (c) and (d) are similar to panels (a) and (b), respectively, but for
fire carbon emissions.


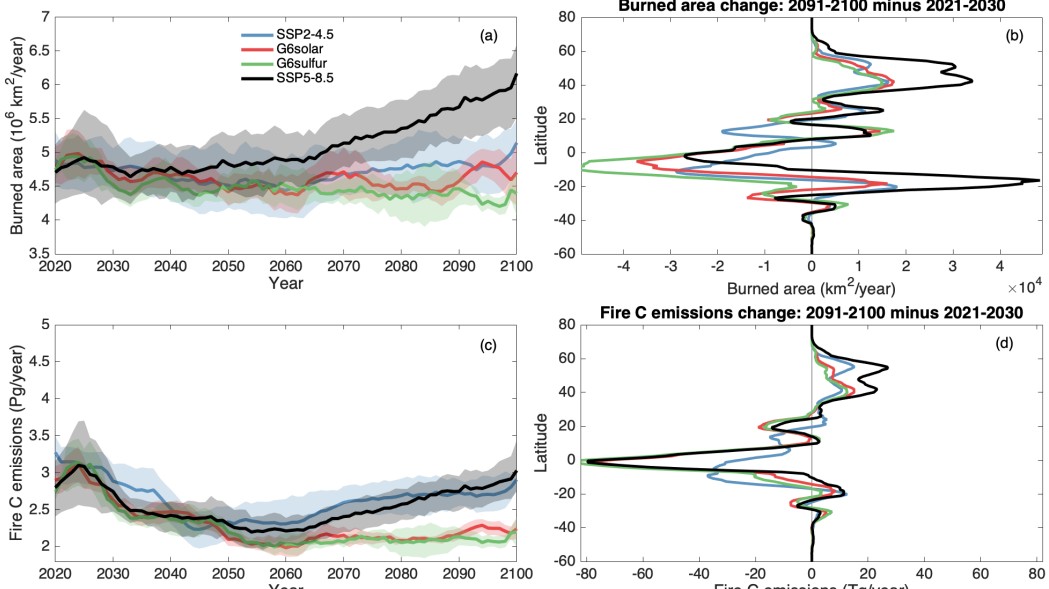

**Figure 2.** Overall global burned area and fire carbon emission trends and changes under the
G6sulfur and G6solar geoengineering scenarios relative to SSP2-4.5 and SSP5-8.5. (a) Time series
of global burned area from 2020 to 2100 under the G6sulfur, G6solar, SSP2-4.5, and SSP5-8.5
scenarios (represented by different colors). For the scenarios with multiple simulations, the ranges
are also shown by the shaded areas. The time series are shown as 5-year moving averages. (b)
Zonal changes (absolute value) of burned area in the period 2091-2100 relative to the period 2021-
2030 (calculated by the value in 2091-2100 minus the value in 2021-2030), under the G6sulfur,
G6solar, SSP2-4.5, and SSP5-8.5 scenarios (represented by different colors, color code is the same
as it in panel a). 5-degree moving average were applied to the shown zonal changes. Panels (c) and
(d) are similar to panels (a) and (b), respectively, but for fire carbon emissions.




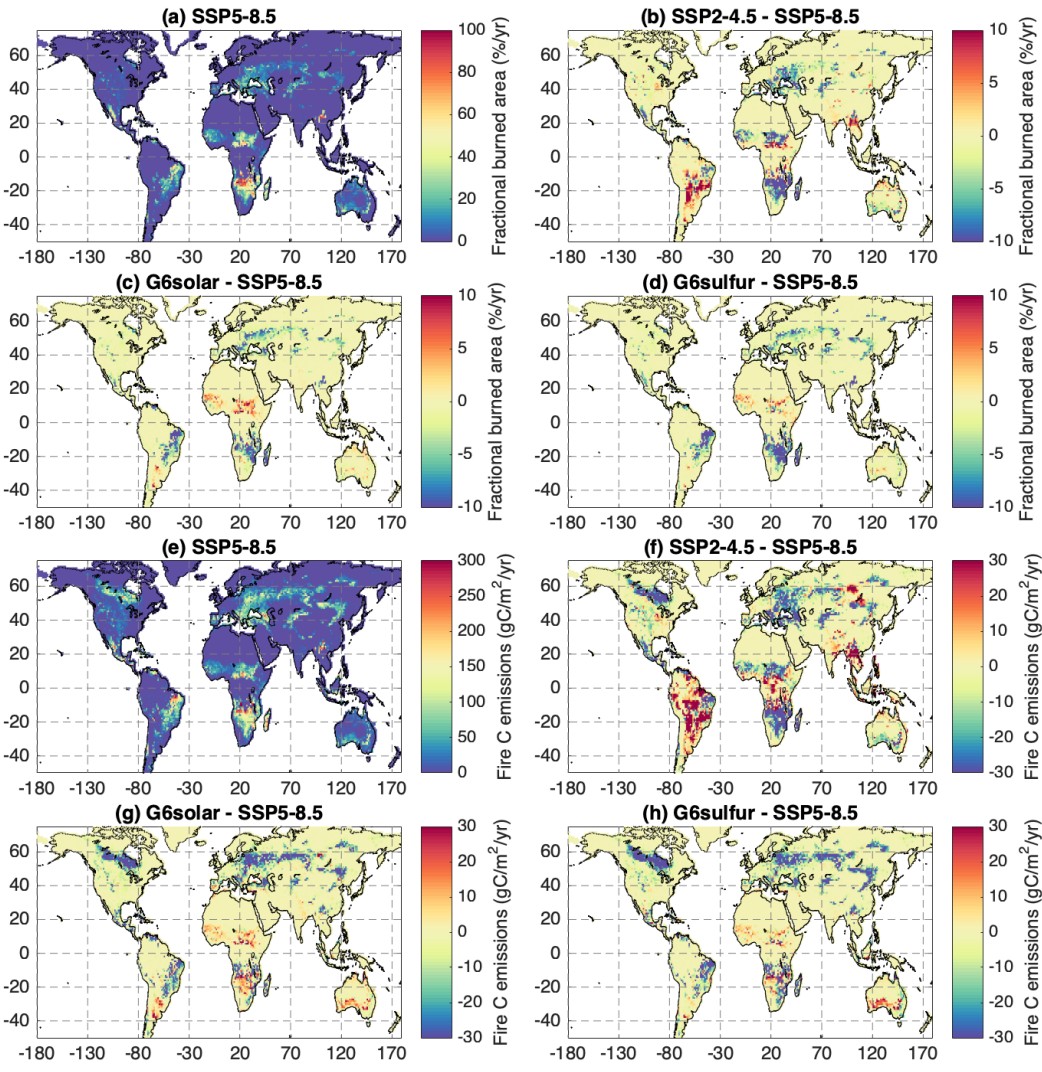

**Figure 3.** Fractional burned area (%/year) and fire carbon missions (gC/m²/year) averaged for
2091-2100. (a) Spatial distribution of fractional burned area (%/year) averaged for 2091-2100
under SSP5-8.5. The difference in surface temperature of (b) SSP2-4.5 from SSP5-8.5 (c) G6Solar
from SSP5-8.5, and (d) G6Sulfur from SSP5-8.5 averaged for 2091-2100. (e-h) are similar to (a-
d) but for fire carbon missions (gC/m²/year). For a scenario with multiple simulations (i.e., SSP5-
8.5, SSP2-4.5, G6Sulfur, and G6Solar), simulation mean is shown.


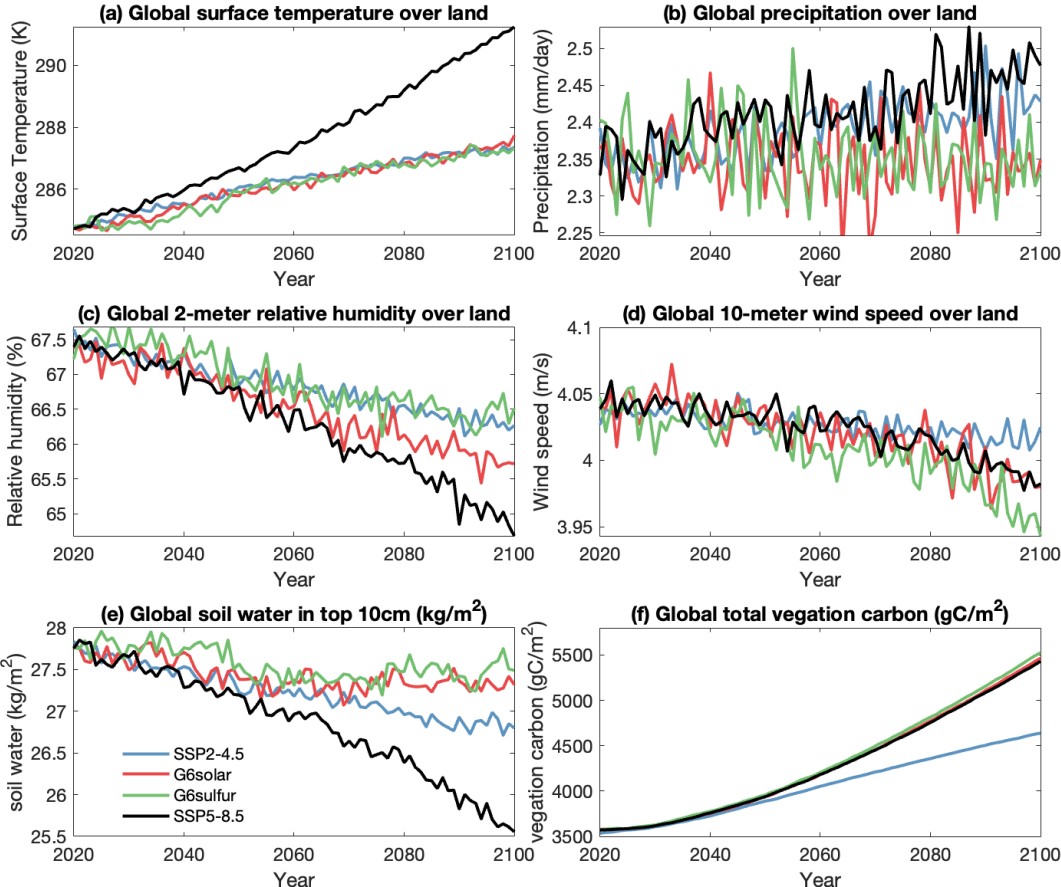

**Figure 4.** Time series of mean (a) surface temperature (K), (b) precipitation (mm/day) over the
land, (c) 2-meter relative humidity (%) over the land, (d) 10-meter wind speed (m/s) over the land,
(e) soil water content at top 10 cm (kg/m$^2$), and (f) vegetation carbon excluding carbon pool
(Gc/m$^2$). For a scenario with multiple simulations (i.e., SSP5-8.5, SSP2-4.5, G6Sulfur, and
G6Solar), simulation means are shown.

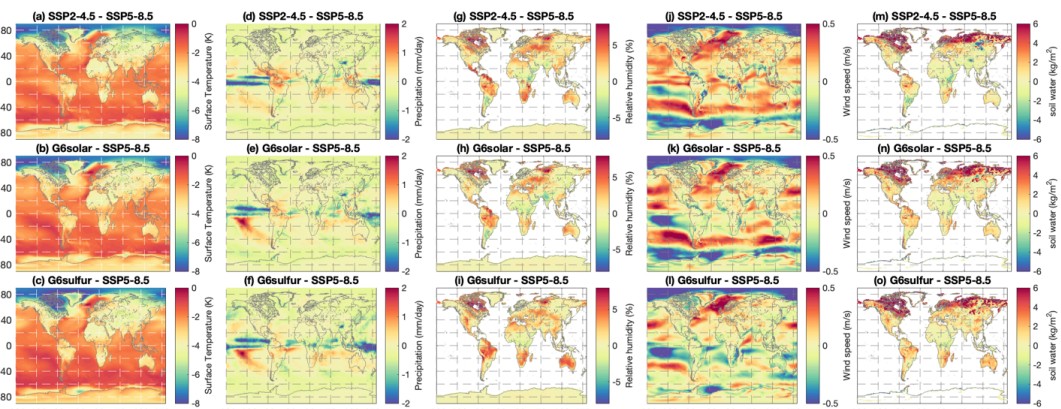

**Figure 5.** The difference in surface temperature (K) of (a) SSP2-4.5 from SSP5-8.5 (b) G6Solar from SSP5-8.5, (c) G6Sulfur from SSP5-8.5. (d-f) are the same as (a-c) but for precipitation (mm/day). (g-i) are the same as (a-c) but for 2-meter relative humidity (%). (j-l) are the same as (a-c) but for 10-meter wind speed (m/s). (m-o) are the same as (a-c) but for soil water content at top 10 cm (kg/m$^2$).


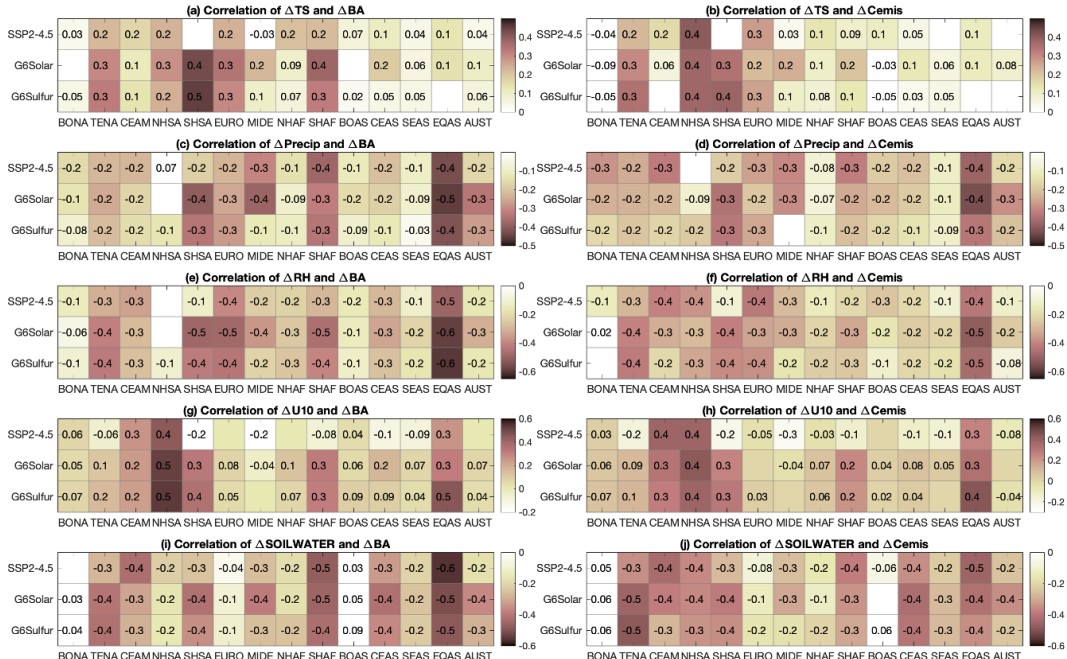


**Figure 6.** Correlations of (a) surface temperature change (ΔTS) and burned area change for SSP2-
4.5, G6Solar, and G6Sulfur, and (b) ΔTS and fire carbon emission change (ΔCemis) for SSP2-4.5,
G6Solar, and G6Sulfur. Only correlations that are significant are labeled (p value <= 0.1). For
SSP2-4.5, ΔTS is calculated for individual model grids within the region and annual values. It is
defined as TS of SSP2-4.5 minus TS of SSP5-8.5 (the reference case). For G6Solar and G6Sulfur,
ΔTS is defined in the same way as SSP2-4.5. ΔBA and ΔCemis are defined in the same way as
ΔTS. (c-d) are the same as (a-b) but for precipitation change (ΔPrecip). (e-f) are the same as (a-b)
but for relative humidity change (ΔRH). (g-h) are the same as (a-b) but for 10-meter wind speed
change (ΔU10). (i-j) are the same as (a-b) but for the change in soil water content at top 10 cm
(ΔSOILWATER). Correlations are calculated for 14 fire regions (x-axis), following Giglio et al.
(2010), namely Boreal North America (BONA), Temperate North America (TENA), Central
America (CEAM), Northern Hemisphere South America (NHSA), Southern Hemisphere South
America (SHSA), Europe (EURO), Middle East (MIDE), Northern Hemisphere Africa (NHAF),
Southern Hemisphere Africa (SHAF), Boreal Asia (BOAS), Central Asia (CEAS), Southeast Asia
(SEAS), Equatorial Asia (EQAS), and Australia and New Zealand (AUST). The definition of the
regions can be found in Figure S3.



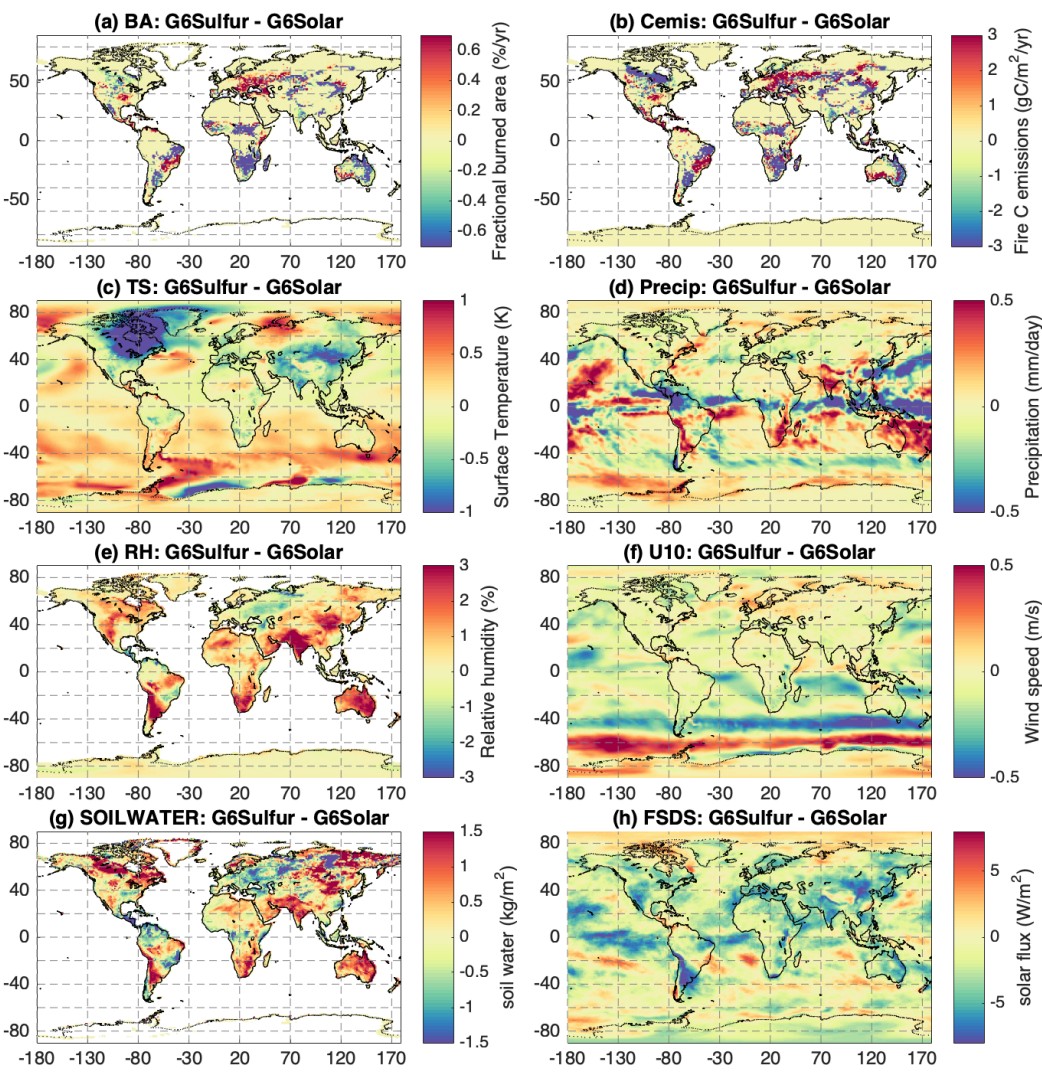

**Figure 7**. The difference between G6Sulfur and G6Solar in (a) burned area fraction (BA; %/yr), (b) fire carbon emissions (Cemis; $gC/m^2/yr$), (c) surface temperature (TS; K), (d) precipitation (Precip; mm/day), (e) 2-meter relative humidity (RH; %), (f) 10-meter wind speed (U10; m/s), (g) soil water content at top 10 cm (Soilwater; $kg/m^2$), and (h) downwelling solar flux at the surface (FSDS; $W/m^2$) averaged for 2091-2100.

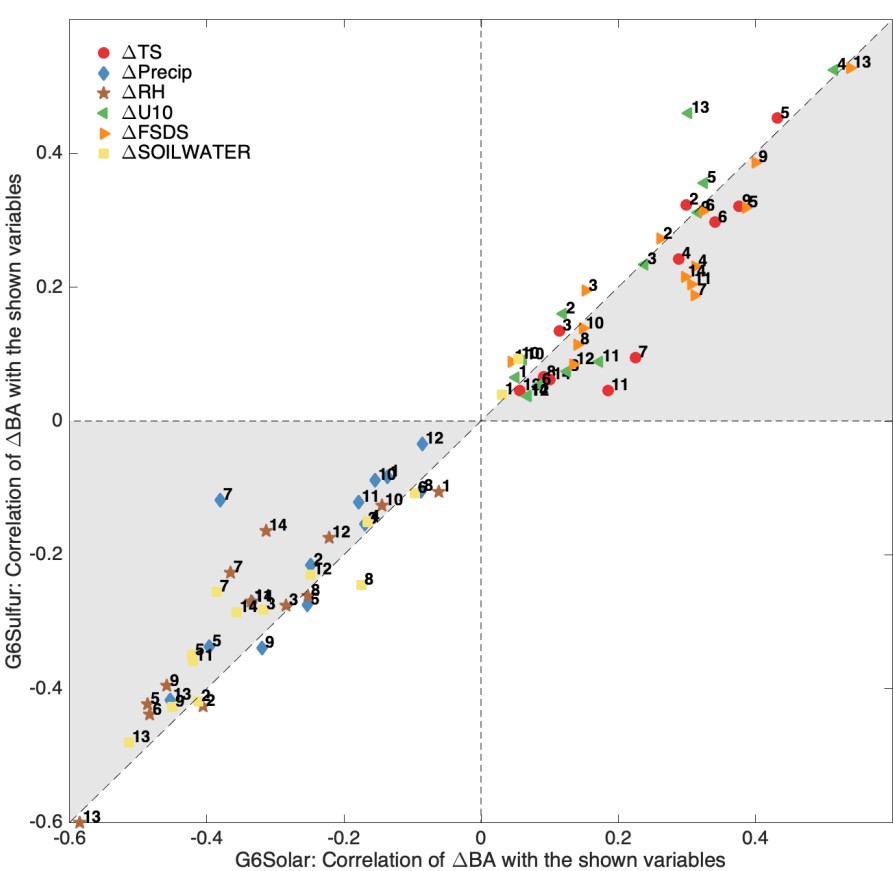

**Figure 8.** Correlations between burned area change in G6Solar from SSP5-8.5 (ΔBA) with the change in other variables in G6Solar from SSP5-8.5 (x-axis) versus correlations between burned area change in G6Solar from SSP5-8.5 (ΔBA) with the change in other variables in G6Sulfur from SSP5-8.5 (y-axis). The variables shown here are surface temperature change (ΔTS), precipitation change (ΔPrecip), 2-meter relative humidity change (ΔRH), 10-meter wind speed change (ΔU10), soil water content in top 10 cm change (ΔSOILWATER), and downwelling solar flux at the surface change (ΔFSDS). The numbers labeled in the figure correspond to the region: 1–Boreal North America, 2–Temperate North America, 3–Central America, 4–Northern Hemisphere South America, 5–Southern Hemisphere South America, 6–Europe, 7–Middle East, 8–Northern Hemisphere Africa, 9–Southern Hemisphere Africa, 10–Boreal Asia, 11–Central Asia, 12–Southeast Asia, 13–Equatorial Asia, and 14–Australia and New Zealand. The definition of the regions can be found in Figure S3. The shade highlights where correlation with ΔBA is larger than correlation with ΔCemis.



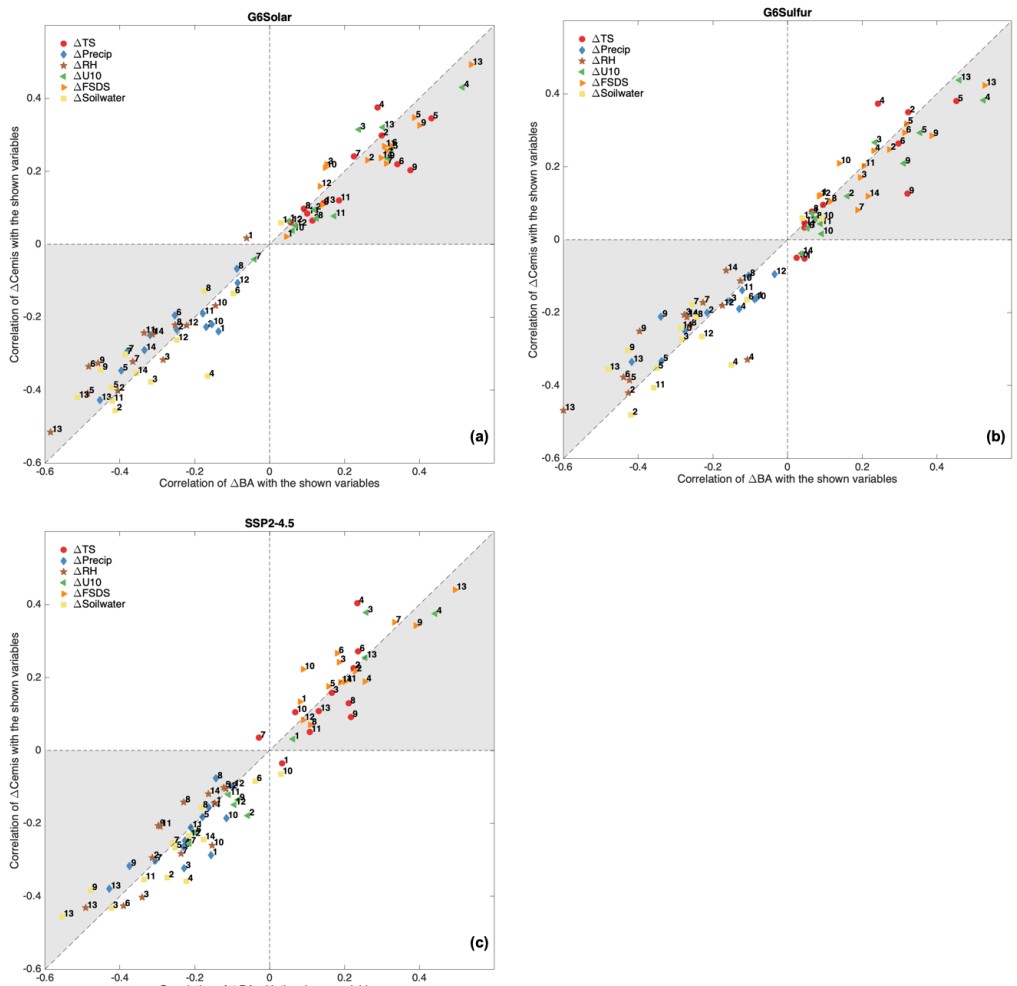

1005

**Figure 9.** (a) Correlations between burned area change in G6Solar from SSP5-8.5 (ΔBA) with the change in other variables in G6Solar from SSP5-8.5 (x-axis) versus correlations between fire carbon emission change in G6Solar from SSP5-8.5 (ΔBA) with the change in other variables in G6Solar from SSP5-8.5 (y-axis). The variables shown here are surface temperature change (ΔTS), precipitation change (ΔPrecip), 2-meter relative humidity change (ΔRH), 10-meter wind speed change (ΔU10), soil water content in top 10 cm change (ΔSOILWATER), and downwelling solar flux at the surface change (ΔFSDS). The numbers labeled in the figure correspond to the region: 1–Boreal North America, 2–Temperate North America, 3–Central America, 4–Northern Hemisphere South America, 5–Southern Hemisphere South America, 6–Europe, 7–Middle East, 8–Northern Hemisphere Africa, 9–Southern Hemisphere Africa, 10–Boreal Asia, 11–Central Asia, 12–Southeast Asia, 13–Equatorial Asia, and 14–Australia and New Zealand. The definition of the regions can be found in Figure S3. The shade highlights where correlation with ΔBA is larger than correlation with ΔCemis. (b) is the same as (a) but for G6Sulfur. (c) is the same as (a) but for SSP2-4.5.