# Peer review of "Impact of Solar Geoengineering on Wildfires in the 21st Century in CESM2/WACCM6"

_Atmospheric Chemistry and Physics, 2022_

## Referee Comment (RC1)

**Peer Review: Tang, et al. 2023, Impact of Solar Geoengineering on Wildfires in the 21st Century in CESM2/WACCM6**

**General Comments**

This study investigates whether and how solar geoengineering affects wildfire impacts in CESM2 simulations by examining the major contributing factors to wildfire activity and how they change in response to geoengineering. The authors find that, in most regions, solar geoengineering causes reductions in wildfire activity, largely due to reduced surface temperatures and increased relative humidity and soil moisture, despite reduced precipitation.

Overall, this is a sound study with meaningful results, and reading it was interesting and informative; wildfires are an important aspect of the climate system, and a study about how solar geoengineering does or does not offset possible changes under global warming is an important contribution to the literature. However, the manuscript needs some work before it can be published. I have two major criticisms. Firstly, the authors need to clarify their methods in some areas; it is not always clear how the authors computed certain numbers presented in their results, and the authors should explain more clearly whether they are averaging over a certain period or how they accounted for certain factors when computing statistics. Secondly, many of the figures are unpolished or incomplete, with missing uncertainties, ensemble spread, or statistical significance; inconsistent color schemes; frustrating or confusing color scales; and captions that do not fully explain how data are calculated. Correcting these issues would improve the clarity, ease of reading, and repeatability of the study. Specific comments and technical corrections are listed below, and I recommend that the study be accepted with minor revisions so that these findings can be published.

**Specific Comments**

Abstract

- Lines 37-38: "a global reduction in burned area and fire carbon emissions" is somewhat misleading; solar geoengineering reduces these relative to SSP5-8.5, but burned area does not appear to be meaningfully reduced relative to present day

1. Introduction

- Lines 92-94: I was unable to find in Robock (2020) where he says that controlling for temperature overcompensates for changes in the hydrological cycle; he just mentions

that GLENS "does not show that precipitation and temperature can be controlled at the same time". The following might be better citations here:

- ○ Lee, et al. (2020): Expanding the design space of stratospheric aerosol geoengineering to include precipitation-based metrics and explore trade-offs, Earth System Dynamics, doi:10.5194/esd-11-1051-2020
- ○ Tilmes, et al. (2013): The hydrological impact of geoengineering in the Geoengineering Model Intercomparison Project (GeoMIP), JGR Atmospheres, doi:10.1002/jgrd.50868
- ○ Bala, et al. (2008): Impact of geoengineering schemes on the global hydrological cycle, PNAS, doi:10.1073/pnas.0711648105

- Lines 101-102: You should elaborate on how important this feedback is (or isn't), so readers unfamiliar with fire schemes can get a sense of how meaningful the results are, and how much an improvement to the model to include this feedback would (or wouldn't) improve your results

**2. Model Description**

- Lines 128-129: "Agricultural fires" refers specifically to fires intentionally set for agricultural reasons, correct? If deforestation fires are also intentional, why are they included?
- Lines 186-188: This is globally-averaged forcing, correct? Additionally, the wording "going from 8.5 $W/m^2$ to 4.5 $W/m^2$ by 2100" is a bit confusing. I would clarify - "Both of these geoengineering scenarios aim to reduce globally-averaged forcing from the ScenarioMIP Tier 1 high-forcing scenario (SSP5-8.5), which averages 8.5 $W/m^2$ of forcing by 2100, to the medium-forcing scenario (SSP2-4.5), which averages 4.5 $W/m^2$ of forcing by 2100." Additionally, you should clarify whether the goal of each experiment was to match the forcing directly (e.g., 8.5 $W/m^2 \rightarrow$ 4.5 $W/m^2$) or to match the surface temperature (e.g., surface temperature of 8.5 $W/m^2 \rightarrow$ surface temperature of 4.5 $W/m^2$).
- Lines 194-196: Can you elaborate on the feedback algorithm at all, for the benefit of those unfamiliar with it? Even one more sentence, such as "The feedback algorithm chooses the reduction in solar constant or $SO_2$ injection quantity based on the prescribed goals, and it adjusts this quantity each year to correct for differences between the simulated climate state and the target" would be helpful.
- Lines 201-202: I was looking for the horizontal resolution back in 2.1, where you gave the vertical resolution of WACCM; consider moving it there

- Lines 203-204: Whenever you direct someone to the supplementary, I recommend including exactly what they should expect to find. Right now you have "see Table S1"; I suggest changing to "see Table S1 for ensemble sizes" or similar

**3. Results**
- Line 211: I'm guessing that all your numbers in parentheses are averages for the 2091-2100 period relative to the 2021-2030 period. You should say this explicitly
- Lines 215-216: It would be helpful if you could include some context about why burned area is expected to decrease in some regions under some scenarios; this is certainly explored in-depth later in the paper, but for a first-time reader, one sentence here - "while total burned area is expected to increase under most global warming scenarios, burned area may decrease in some regions due to reduced 2m relative humidity and reduced soil moisture" or similar - would be very helpful.
- Line 231: Do the ranges represent ensemble spread? You should clarify your methods explicitly

**4. Mechanisms**
- Lines 296-297: "The correlations calculated here account for spatial variability within the region and interannual variability during 2091-2100" How, exactly? Please say explicitly what you did to account for this
- Lines 307-308: "This suggests that the changes in area burnt in these regions are not predominantly driven by the surface temperature changes, but by other factors" This seems like an odd speculation to make, given that the purpose of your study is to determine how geoengineering affects fires - do your results support this, or not? This also contradicts what you say in the discussion, which is that these regions are not very sensitive to any of the factors considered in this study
- Lines 349-352: If you're going to discuss this analysis, you should provide the actual numbers, either here or in the supplementary
- Lines 371-374: Does the model output support this? It should be relatively easy to compute evapotranspiration and confirm or deny this
- Lines 465-468: I disagree with this sentence; G6Solar does not provide more "direct" climate impacts than G6Sulfur, and it does not follow that G6Solar would therefore be expected to have larger impacts, as both experiments have similar reductions in downward forcing and surface temperature

- Lines 468-470: Without shading or error bars in the figure, there is no way to know whether or not the differences between G6Solar and G6Sulfur are significant

5. Conclusions
- Line 535-537: Be careful with wording; burned area decreases in the geoengineering scenarios relative to SSP5-8.5, but there is no decrease relative to present day, or SSP2-4.5, that is statistically significant in Fig. 1

Figures
- Figures 1, 2, and 4: I would like to see uncertainty better represented in these figures. Ensemble spread is missing for SSP2-4.5 in Fig. 1a and 1c, for SSP2-4.5 and SSP5-8.5 in Fig. 1b and 1d, for all simulations in Fig. 2b and 2d, and everywhere in Fig. 4. For time series, I would like to at least see the ensemble spread; ideally, the uncertainty introduced by taking the 5-year running average would also be accounted for. For plots showing a difference between two time periods, the uncertainty for both the experimental period and the reference period should be reflected.
- Figure 1: I suggest flipping the axes for panels b and d (latitude, the independent variable, on the horizontal axis; burned area or C emissions, the dependent variables, on the vertical axis).
- Figure 1: The "$10^4$" for panel b (At least, I think it belongs to panel b and not d) is very out of the way - I suggest making it easier to find, perhaps by including it in the axis description: "Burned Area ($10^4$ km$^2$/yr)"
- Figure 1: According to Table S1, SSP2-4.5 also has multiple ensemble members; why isn't there shading to show the ensemble spread like there is for SSP5-8.5? Why isn't there shading in panels b and d at all?
- Figure 2: Same comment as above about $10^4$ for panel b
- Figure 2: The color scheme should be consistent with Figure 1. SSP2-4.5 and SSP5-8.5 were red and purple in Fig. 1 and they're blue and black in Fig. 2. Using the same colors for each run throughout the paper would be very helpful
- Figures 3, 5, and 7: I would like to see statistical significance on all of these maps, i.e., hatching or shading where the changes are not significant.
- Figure 4: same comment about color scheme consistency across figures
- Figure 5: for the temperature plots, I strongly recommend a different color scheme. For every other plot in this figure, yellow represents no change, but on the temperature plots,

both red and blue represent cooling, and yellow indicates moderate cooling. This is quite confusing; I suggest either changing the limits of the color scheme from +8 to -8 to match the other plots, with red for warming, yellow for no change, and blue for cooling (even though there's no warming), or keep it from 0 to -8 and have the color scale just be yellow and blue. Either way, I advise no red on the temperature plots if there's no warming!

- Figure 5: for precipitation, humidity, and soil moisture plots, by convention, red is usually used for drier conditions and blue for wetter conditions. I recommend you flip the color scale
- Figure 6: This plot is very hard to read, for two reasons: firstly, all the panels have different color scales, and secondly, the different shades of yellow and orange blend together and make it hard to search for a specific result. I recommend you change the color scheme to be more like Figure S5; that plot uses a consistent color scheme for the entire figure, and the contrasting blues and reds make it much easier to identify patterns.
- Figure 7: see previous comments on plots in Fig. 5; here, red = warming and blue = cooling, and this one is much easier to interpret. Same comments about the hydrological plots
- Figures 8-9: I assume that all "changes" refer to 2091-2100 averages? You should say this explicitly in the caption
- Figures 8-9: Some of the number labels are covered up by other data points; can you fix this?

**Technical Corrections**
- Lines 183-196: This could probably all be one paragraph
- Lines 223-224: "the fire carbon emissions and burned area generally show trends consistent with burned area" I think there's a typo in here
- Line 470: There is no Figure 4g; I assume you mean 4f?
- The "Conclusions" section is mistakenly numbered "4" instead of "5"
- Figure 8: I think there is a typo in the caption, line 994; I believe "G6Solar" should be "G6Sulfur"

---

## Author Comment (AC1)

**Reviewer 1:**

This study investigates whether and how solar geoengineering affects wildfire impacts in CESM2 simulations by examining the major contributing factors to wildfire activity and how they change in response to geoengineering. The authors find that, in most regions, solar geoengineering causes reductions in wildfire activity, largely due to reduced surface temperatures and increased relative humidity and soil moisture, despite reduced precipitation.

Overall, this is a sound study with meaningful results, and reading it was interesting and informative; wildfires are an important aspect of the climate system, and a study about how solar geoengineering does or does not offset possible changes under global warming is an important contribution to the literature. However, the manuscript needs some work before it can be published. I have two major criticisms.

Firstly, the authors need to clarify their methods in some areas; it is not always clear how the authors computed certain numbers presented in their results, and the authors should explain more clearly whether they are averaging over a certain period or how they accounted for certain factors when computing statistics.

**Response:** Thank you. We have revised the manuscript to provide details and clarifications on the methods. Please see the response to the specific comments below for details.

Secondly, many of the figures are unpolished or incomplete, with missing uncertainties, ensemble spread, or statistical significance; inconsistent color schemes; frustrating or confusing color scales; and captions that do not fully explain how data are calculated. Correcting these issues would improve the clarity, ease of reading, and repeatability of the study. Specific comments and technical corrections are provided in the supplement, and I recommend that the study be accepted with minor revisions so that these findings can be published.

**Response:** We have revised the figures and addressed the issues pointed out by the reviewers. Please see the response to the specific comments below for details.

**Specific Comments**

**Abstract**

1. Lines 37-38: "a global reduction in burned area and fire carbon emissions" is somewhat misleading; solar geoengineering reduces these relative to SSP5-8.5, but burned area does not appear to be meaningfully reduced relative to present day

**Response:** Thank you for pointing this out. We have changed "a global reduction in burned area and fire carbon emissions" to "a global reduction in burned area and fire carbon emissions by the end of the century relative to their base-climate scenario SSP5-8.5".

**Introduction**

2. Lines 92-94: I was unable to find in Robock (2020) where he says that controlling for temperature overcompensates for changes in the hydrological cycle; he just mentions that GLENS "does not show that precipitation and temperature can be controlled at the same time". The following might be better citations here:

○ Lee, et al. (2020): Expanding the design space of stratospheric aerosol geoengineering to include precipitation-based metrics and explore trade-offs, Earth System Dynamics, doi:10.5194/esd-11-1051-2020.
○ Tilmes, et al. (2013): The hydrological impact of geoengineering in the Geoengineering Model Intercomparison Project (GeoMIP), JGR Atmospheres, doi:10.1002/jgrd.50868
○ Bala, et al. (2008): Impact of geoengineering schemes on the global hydrological cycle, PNAS, doi:10.1073/pnas.0711648105

**Response:** Thank you for the suggestion. We have changed the reference "Robock (2020)" to "Bala, et al. (2008), Tilmes, et al. (2013), and Lee, et al. (2020)".

3. Lines 101-102: You should elaborate on how important this feedback is (or isn't), so readers unfamiliar with fire schemes can get a sense of how meaningful the results are, and how much an improvement to the model to include this feedback would (or wouldn't) improve your results

**Response:** We added the following statement in the introduction:

"A coupling of fire emissions to the atmosphere would allow to identify additional climate feedback including changes to climate and the vegetation."

And we added the following statement to section 2.1 where we describe the fire scheme in CESM2/CLM5:

"Changes in fires can have an impact on radiation, precipitation, and therefore vegetation. However, since this paper mainly focuses on the impacts of solar geoengineering on wildfires instead of the other way around, we do not expect the uncoupled fire emissions to have a large impact on our results, but future studies will be needed to further understand the impact."

**Model Description**

4. Lines 128-129: "Agricultural fires" refers specifically to fires intentionally set for agricultural reasons, correct? If deforestation fires are also intentional, why are they included?

**Response:** Agricultural fires in the fire scheme in CESM2/CLM5 only refers to fires in cropland and does not include deforestation fires.

5. Lines 186-188: This is globally-averaged forcing, correct? Additionally, the wording "going from 8.5 W/m2 to 4.5 W/m2 by 2100" is a bit confusing. I would clarify - "Both of these geoengineering scenarios aim to reduce globally-averaged forcing from the ScenarioMIP Tier 1 high-forcing scenario (SSP5-8.5), which averages 8.5 W/m2 of forcing by 2100, to the medium-forcing scenario (SSP2-4.5), which averages 4.5 W/m2 of forcing by 2100." Additionally, you should clarify whether the goal of each experiment was to match the forcing directly (e.g., 8.5 W/m2 → 4.5 W/m2) or to match the surface temperature (e.g., surface temperature of 8.5 W/m2 → surface temperature of 4.5 W/m2).

**Response:** Thank you. We changed

"Both of these geoengineering scenarios aim to reduce forcing from ScenarioMIP Tier 1 high forcing scenario (SSP5-8.5) to the medium forcing scenario (SSP2-4.5), going from 8.5 to 4.5 $Wm^{-2}$ in 2100."

to "Both of these geoengineering scenarios aim to reduce globally-averaged forcing from the ScenarioMIP Tier 1 high-forcing scenario (SSP5-8.5), which averages 8.5 $W/m^2$ of forcing by 2100, to the medium-forcing scenario (SSP2-4.5), which averages 4.5 $W/m^2$ of forcing by 2100. The geoengineering scenarios were designed to match the surface temperature of SSP2-4.5."

6. Lines 194-196: Can you elaborate on the feedback algorithm at all, for the benefit of those unfamiliar with it? Even one more sentence, such as "The feedback algorithm chooses the reduction in solar constant or SO2 injection quantity based on the prescribed goals, and it adjusts this quantity each year to correct for differences between the simulated climate state and the target" would be helpful.

**Response:** Thank you. We added the following statement to the text:

"The feedback algorithm identifies differences in the global mean surface temperature between the simulated and the prescribed target temperature each year and from that calculates required changes in the solar constant or $SO_2$ injections."

7. Lines 201-202: I was looking for the horizontal resolution back in 2.1, where you gave the vertical resolution of WACCM; consider moving it there

**Response:** We added the following statement to Section 2.1:

"The default horizontal resolution of WACCM6 is $1.25° \times 0.9°$ (longitude $\times$ latitude)."

8. Lines 203-204: Whenever you direct someone to the supplementary, I recommend including exactly what they should expect to find. Right now you have "see Table S1"; I suggest changing to "see Table S1 for ensemble sizes" or similar

**Response:** Thank you. We changed "see Table S1" to "see Table S1 for ensemble sizes". We also changed "see Table S2" to "see Table S2 for projected regional and global change of burned area and fire carbon emissions in 2091-2100 relative to 2021-2030 (%) under different scenarios", and added "see Table S3 for averages of regional and global annual projected burned area (Mha/year) and fire carbon emissions in 2091-2100 under different scenarios" in Section 3.2.

**Results**

9. Line 211: I'm guessing that all your numbers in parentheses are averages for the 2091-2100 period relative to the 2021-2030 period. You should say this explicitly

**Response:** We changed "The largest increases in the global burned area are seen in the SSP5-8.5 scenarios (~20%) and SSP3-7.0 (~10%)." to "The largest increases (averages for the 2091-2100 period relative to the 2021-2030 period) in the global burned area are seen in the SSP5-8.5 scenarios (~20%) and SSP3-7.0 (~10%)."

10. Lines 215-216: It would be helpful if you could include some context about why burned area is expected to decrease in some regions under some scenarios; this is certainly explored in-depth later in the paper, but for a first-time reader, one sentence here - "while total burned area is expected to increase under most global warming scenarios, burned area may decrease in some regions due to reduced 2m relative humidity and reduced soil moisture" or similar - would be very helpful.

**Response:** We added the following statement to Section 3.1:

"While global total burned area is expected to increase under most global warming scenarios, burned area may decrease in some regions due to changes in anthropogenic activities or reduced 2m relative humidity and/or reduced soil moisture."

11. Line 231: Do the ranges represent ensemble spread? You should clarify your methods Explicitly.

**Response:** Thank you. We changed

"The change of the two geoengineering scenarios compared to SSP2-4.5 in the last decade of the century is small in burned area (-2% – -12%) but relatively large in fire carbon emissions (-18% – -23%)."

to

"The change of the two geoengineering scenarios compared to SSP2-4.5 in the last decade of the century is small in burned area (-2% for G6Solar and -12% for G6Sulfur) but relatively large in fire carbon emissions (-18% for G6Solar and -23% for G6Sulfur)."

Other similar sentences in the section have also been changed.

12. Lines 296-297: "The correlations calculated here account for spatial variability within the region and interannual variability during 2091-2100" How, exactly? Please say explicitly what you did to account for this

**Response:** We added the following statement to Section 4.1 to demonstrate the method:

"For example, if a region consists of 500 individual model grids, as we use 10 years of annual data, there will be 5000 (500 × 10) pairs of $\Delta TS$ and $\Delta BA$ to calculate correlations."

13. Lines 307-308: "This suggests that the changes in area burnt in these regions are not predominantly driven by the surface temperature changes, but by other factors" This seems like an odd speculation to make, given that the purpose of your study is to determine how geoengineering affects fires - do your results support this, or not? This also contradicts what you say in the discussion, which is that these regions are not very sensitive to any of the factors considered in this study

**Response:** The purpose of this study is to determine how geoengineering affects fires, however geoengineering can impact fires through factors other than surface temperature changes. Here we found that burned areas and fire emissions are changed under geoengineering, however the impacts of surface temperature change over boreal regions are relatively small, therefore we hypothesize that they may be driven by other factors driven by geoengineering (e.g., hydrological cycle).

To clarify, we changed the statement

"This suggests that the changes in area burnt in these regions are not predominantly driven by the surface temperature changes, but by other factors."

to

"This suggests that the changes in area burnt in these regions might be predominantly driven by other factors changed by geoengineering (e.g., hydrological cycle) rather than the surface temperature changes, which will be analyzed in the following sub-sections."

14. Lines 349-352: If you're going to discuss this analysis, you should provide the actual numbers, either here or in the supplementary

**Response:** We added the values in Supplement (Table S4) and referred it in the main text.

**Table S4**. 1-year lag correlations of precipitation change and burned area change and fire carbon emission change for SSP2-4.5, G6Solar, and G6Sulfur from SSP5-8.5.

|  | Burned area | | | Fire carbon emissions | | |
|---|---|---|---|---|---|---|
|  | SSP2-4.5 | G6Solar | G6Sulfur | SSP2-4.5 | G6Solar | G6Sulfur |

| | | | | | | |
|---|---|---|---|---|---|---|
| BONA (Boreal North America) | -0.12 | -0.09 | -0.06 | -0.20 | -0.15 | -0.10 |
| TENA (Temperate North America) | 0.04 | / | -0.05 | 0.04 | / | / |
| CEAM (Central America) | -0.21 | -0.15 | / | -0.28 | -0.19 | / |
| NHSA (Northern Hemisphere South America) | / | / | -0.18 | / | -0.11 | -0.29 |
| SHSA (Southern Hemisphere South America) | -0.11 | -0.31 | -0.25 | -0.08 | -0.24 | -0.21 |
| EURO (Europe) | -0.11 | -0.15 | 0.07 | -0.06 | -0.04 | 0.11 |
| MIDE (Middle East) | -0.21 | -0.11 | 0.06 | -0.21 | / | 0.15 |
| NHAF (Northern Hemisphere Africa) | -0.11 | -0.07 | -0.06 | -0.05 | -0.05 | -0.05 |
| SHAF (Southern Hemisphere Africa) | -0.26 | -0.13 | -0.20 | -0.19 | -0.03 | -0.07 |
| BOAS (Boreal Asia) | -0.06 | -0.13 | -0.06 | -0.08 | -0.18 | -0.07 |
| CEAS (Central Asia) | -0.10 | / | 0.05 | -0.06 | 0.03 | 0.06 |
| SEAS (Southeast Asia) | / | -0.07 | / | / | -0.07 | / |
| EQAS (Equatorial Asia) | 0.14 | / | 0.11 | 0.17 | / | 0.17 |
| AUST (Australia and New Zealand) | 0.05 | -0.07 | -0.03 | 0.05 | -0.06 | / |

15. Lines 371-374: Does the model output support this? It should be relatively easy to compute evapotranspiration and confirm or deny this
**Response:** Thank you. We checked the model evapotranspiration and the model results do not support this statement (see below). Therefore, we deleted it from the text.

[Figure]

16. Lines 465-468: I disagree with this sentence; G6Solar does not provide more "direct" climate impacts than G6Sulfur, and it does not follow that G6Solar would therefore be expected to have larger impacts, as both experiments have similar reductions in downward forcing and surface temperature
**Response:** We changed
"This is expected since the climate impacts of solar irradiance reduction (G6Solar) is more direct than that of stratospheric sulfate aerosols (G6Sulfur) and stratospheric sulfate aerosols can yield to additional changes (such as higher diffuse radiation that benefits plant growth)."
to

"It is possible that stratospheric sulfate aerosols could yield to additional changes such as higher diffuse radiation that benefits plant growth, which reduces the correlations of the analyzed factors with fires."

17. Lines 468-470: Without shading or error bars in the figure, there is no way to know whether or not the differences between G6Solar and G6Sulfur are significant

**Response:** We deleted the statement.

**Conclusions**

18. Line 535-537: Be careful with wording; burned area decreases in the geoengineering scenarios relative to SSP5-8.5, but there is no decrease relative to present day, or SSP2-4.5, that is statistically significant in Fig. 1

**Response:** We changed the statement to "The global total wildfire burned area is projected to increase under the unmitigated scenario (SSP5-8.5), and decrease under the two geoengineering scenarios (solar irradiance reduction and stratospheric sulfate aerosols) comparing the averages of 2091-2100 relative to 2021-2030."

**Figures**

19. Figures 1, 2, and 4: I would like to see uncertainty better represented in these figures. Ensemble spread is missing for SSP2-4.5 in Fig. 1a and 1c, for SSP2-4.5 and SSP5-8.5 in Fig. 1b and 1d, for all simulations in Fig. 2b and 2d, and everywhere in Fig. 4. For time series, I would like to at least see the ensemble spread; ideally, the uncertainty introduced by taking the 5-year running average would also be accounted for. For plots showing a difference between two time periods, the uncertainty for both the experimental period and the reference period should be reflected.

**Response:** In this study, different scenarios have different numbers of ensemble sizes. Reviewer 2 was concerned that different ensemble sizes may result in biases in the ensemble ranges, which may be misleading. I.e., a scenario with larger ensemble spread may be due to larger ensemble size rather than larger variabilities. In addition, in Section 4 (mechanism of geoengineering impacting fires), we only used the ensemble mean values and the analyses does not involve ensemble spread. Therefore, to be consistent among scenarios and avoid the confusion, we changed the Section 3 and corresponding figures (Figures 1, 2, and S4) and tables (Tables 2 and 3) to only show ensemble mean values rather than spread. Please see the updated manuscript for more information.

[Figure]

**Figure 1.** Overall global burned area and fire carbon emission trends and changes under SSP scenarios. (a) Time series of global burned area from 2020 to 2100 under the SSP1-2.6, SSP2-4.5, SSP3-7.0, and SSP5-8.5 scenarios (represented by different colors). The time series are shown as 5-year moving averages. (b) Zonal changes (absolute value) of burned area in the period 2091-2100 relative to the period 2021-2030 (calculated by the value in 2091-2100 minus the value in 2021-2030), under the SSP1-2.6, SSP2-4.5, SSP3-7.0, and SSP5-8.5 scenarios (represented by different colors, color code is the same as it in panel a). 5-degree moving average were applied to the shown zonal changes. Panels (c) and (d) are similar to panels (a) and (b), respectively, but for fire carbon emissions.

[Figure]

**Figure 2.** Overall global burned area and fire carbon emission trends and changes under the G6sulfur and G6solar geoengineering scenarios relative to SSP2-4.5 and SSP5-8.5. (a) Time series of global burned area from 2020 to 2100 under the G6sulfur, G6solar, SSP2-4.5, and SSP5-8.5 scenarios (represented by different colors). The time series are shown as 5-year moving averages. (b) Zonal changes (absolute value) of burned area in the period 2091-2100 relative to the period 2021-2030 (calculated by the value in 2091-2100 minus the value in 2021-2030), under the G6sulfur, G6solar, SSP2-4.5, and SSP5-8.5 scenarios (represented by different colors, color code is the same as it in panel a). 5-degree moving average were applied to the shown zonal changes. Panels (c) and (d) are similar to panels (a) and (b), respectively, but for fire carbon emissions.

[Figure]

**Figure 4.** Time series of mean (a) surface temperature (K), (b) precipitation (mm/day) over the land, (c) 2-meter relative humidity (%) over the land, (d) 10-meter wind speed (m/s) over the land, (e) soil water content at top 10 cm (kg/m$^2$), and (f) vegetation carbon excluding carbon pool (Gc/m$^2$). For a scenario with multiple simulations (i.e., SSP5-8.5, SSP2-4.5, G6Sulfur, and G6Solar), simulation means are shown.

20. Figure 1: I suggest flipping the axes for panels b and d (latitude, the independent variable, on the horizontal axis; burned area or C emissions, the dependent variables, on the vertical axis).
**Response:** We flipped the axes for panels b and d in Figures 1 and 2. Please see the updated figures in Response to Comment #19.

21. Figure 1: The "10^4" for panel b (At least, I think it belongs to panel b and not d) is very out of the way - I suggest making it easier to find, perhaps by including it in the axis description: "Burned Area (10^4 km2/yr)"
**Response:** We revised the axis description in Figures 1 and 2. Please see the updated figures in Response to Comment #19.

22. Figure 1: According to Table S1, SSP2-4.5 also has multiple ensemble members; why isn't there shading to show the ensemble spread like there is for SSP5-8.5? Why isn't there shading in panels b and d at all?
**Response:** We removed ensemble spread from all figures. Please see the updated figures in Response to Comment #19.

23. Figure 2: Same comment as above about 10^4 for panel b
**Response:** We revised the axis description. Please Please see the updated figures in Response to Comment #19.

24. Figure 2: The color scheme should be consistent with Figure 1. SSP2-4.5 and SSP5-8.5 were red and purple in Fig. 1 and they're blue and black in Fig. 2. Using the same colors for each run throughout the paper would be very helpful
**Response:** We thank the reviewer for pointing this out. We have revised Figures 1, 2, and 4 to use the consistent and colorblind friendly colors. Please see the updated figures in Response to Comment #19.

25. Figures 3, 5, and 7: I would like to see statistical significance on all of these maps, i.e., hatching or shading where the changes are not significant.
**Response:** Figure 3 is updated to only show results where fractional burned area or fire carbon emissions not equals to 0. Figures 5 and 7 are updated with insignificant points marked on the maps. Please see the updated figure below.

[Figure]

**Figure 3.** Fractional burned area (%/year) and fire carbon missions (gC/m$^2$/year) averaged for 2091-2100. (a) Spatial distribution of fractional burned area (%/year) averaged for 2091-2100 under SSP5-8.5. Results are not shown for model grids where fractional burned area equals to 0. The difference in fractional burned area of (b) SSP2-4.5 from SSP5-8.5 (c) G6Solar from SSP5-8.5, and (d) G6Sulfur from SSP5-8.5 averaged for 2091-2100. Results are not shown for model grids where the difference in fractional burned area equals to 0. (e-h) are similar to (a-d) but for fire carbon missions (gC/m$^2$/year). For a scenario with multiple simulations (i.e., SSP5-8.5, SSP2-4.5, G6Sulfur, and G6Solar), simulation mean is shown.

[Figure]

**Figure 5.** The difference in surface temperature (K) of (a) SSP2-4.5 from SSP5-8.5 (b) G6Solar from SSP5-8.5, (c) G6Sulfur from SSP5-8.5. (d-f) are the same as (a-c) but for precipitation (mm/day). (g-i) are the same as (a-c) but for 2-meter relative humidity (%). (j-l) are the same as (a-c) but for 10-meter wind speed (m/s). (m-o) are the same as (a-c) but for soil water content at top 10 cm (kg/m$^2$). The grids where SSP2-4.5, G6Sulfur, or G6Solar is not significantly different from SSP5-8.5 is marked with white shade. Taking precipitation of SSP2-4.5 as an example, the significance for each model grid is calculated by student t-test (p value is 0.1) using 10 years of SSP2-4.5 precipitation (10 data points) and 10 years of SSP5-8.5 precipitation (10 data points).

[Figure]

**Figure 7.** The difference between G6Sulfur and G6Solar in (a) burned area fraction (BA; %/yr), (b) fire carbon emissions (Cemis; gC/m$^2$/yr), (c) surface temperature (TS; K), (d) precipitation (Precip; mm/day), (e) 2-meter relative humidity (RH; %), (f) 10-meter wind speed (U10; m/s), (g) soil water content at top 10 cm (Soilwater; kg/m$^2$), and (h) downwelling solar flux at the surface (FSDS; W/m$^2$) averaged for 2091-2100. The grids where SSP2-4.5, G6Sulfur, or G6Solar is not significantly different from SSP5-8.5 is marked with white shade. Taking precipitation of SSP2-

4.5 as an example, the significance for each model grid is calculated by student t-test (p value is 0.1) using 10 years of SSP2-4.5 precipitation data during 2091-2100 (10 data points) and 10 years of SSP5-8.5 precipitation data during 2091-2100 (10 data points).

26. Figure 4: same comment about color scheme consistency across figures
**Response:** We thank the reviewer for pointing this out. We have revised Figures 1, 2, and 4 to use the consistent and colorblind friendly colors. Please see the updated figures in Response to Comment #19.

27. Figure 5: for the temperature plots, I strongly recommend a different color scheme. For every other plot in this figure, yellow represents no change, but on the temperature plots, both red and blue represent cooling, and yellow indicates moderate cooling. This is quite confusing; I suggest either changing the limits of the color scheme from +8 to -8 to match the other plots, with red for warming, yellow for no change, and blue for cooling (even though there's no warming), or keep it from 0 to -8 and have the color scale just be yellow and blue. Either way, I advise no red on the temperature plots if there's no warming!
**Response:** We changed the color scheme from +8 to -8 to match the other plots. Please see the updated figures in Response to Comment #25.

28. Figure 5: for precipitation, humidity, and soil moisture plots, by convention, red is usually used for drier conditions and blue for wetter conditions. I recommend you flip the color scale
**Response:** We flip the color scale for precipitation, humidity, and soil moisture plots. Please see the updated figures in Response to Comment #25.

29. Figure 6: This plot is very hard to read, for two reasons: firstly, all the panels have different color scales, and secondly, the different shades of yellow and orange blend together and make it hard to search for a specific result. I recommend you change the color scheme to be more like Figure S5; that plot uses a consistent color scheme for the entire figure, and the contrasting blues and reds make it much easier to identify patterns.
**Response:** We updated Figure 6 using the same color scale and color scheme. Please see below.

[Figure]

**Figure 6.** Correlations of (a) surface temperature change (ΔTS) and burned area change for SSP2-4.5, G6Solar, and G6Sulfur, and (b) ΔTS and fire carbon emission change (ΔCemis) for SSP2-4.5, G6Solar, and G6Sulfur. Only correlations that are significant are labeled (p value <= 0.1). For SSP2-4.5, ΔTS is calculated for individual model grids within the region and annual values. It is defined as TS of SSP2-4.5 minus TS of SSP5-8.5 (the reference case). For G6Solar and G6Sulfur, ΔTS is defined in the same way as SSP2-4.5. ΔBA and ΔCemis are defined in the same way as ΔTS. (c-d) are the same as (a-b) but for precipitation change (ΔPrecip). (e-f) are the same as (a-b) but for relative humidity change (ΔRH). (g-h) are the same as (a-b) but for 10-meter wind speed change (ΔU10). (i-j) are the same as (a-b) but for the change in soil water content at top 10 cm (ΔSOILWATER). Correlations are calculated for 14 fire regions (x-axis), following Giglio et al. (2010), namely Boreal North America (BONA), Temperate North America (TENA), Central America (CEAM), Northern Hemisphere South America (NHSA), Southern Hemisphere South America (SHSA), Europe (EURO), Middle East (MIDE), Northern Hemisphere Africa (NHAF), Southern Hemisphere Africa (SHAF), Boreal Asia (BOAS), Central Asia (CEAS), Southeast Asia (SEAS), Equatorial Asia (EQAS), and Australia and New Zealand (AUST). The definition of the regions can be found in Figure S3.

30. Figure 7: see previous comments on plots in Fig. 5; here, red = warming and blue = cooling, and this one is much easier to interpret. Same comments about the hydrological plots
**Response:** We flip the color scale for precipitation, humidity, and soil moisture plots. Please see the updated figures in Response to Comment #25.

31. Figures 8-9: I assume that all "changes" refer to 2091-2100 averages? You should say this explicitly in the caption
**Response:** We updated the captions for Figures 8 and 9.

**32. Figures 8-9:** Some of the number labels are covered up by other data points; can you fix this?

**Response:** It is hard to show all the data points and number labels in one figure without overlapping. We added the following figures in the supplement showing each variable separately.

[Figure]

**Figure S13**. Same as Figure 8 but showing each variable separately.

[Figure]

**Figure S14**. Same as Figure 9a but showing each variable separately.

[Figure]

**Figure S15**. Same as Figure 9b but showing each variable separately.

[Figure]

**Figure S16.** Same as Figure 9c but showing each variable separately.

**Technical Corrections**

33. Lines 183-196: This could probably all be one paragraph
**Response:** We merged them to one paragraph.

34. Lines 223-224: "the fire carbon emissions and burned area generally show trends consistent with burned area" I think there's a typo in here
**Response:** We changed "the fire carbon emissions and burned area generally show trends consistent with burned area" to "the fire carbon emissions generally show trends consistent with burned area".

35. Line 470: There is no Figure 4g; I assume you mean 4f?
**Response:** The sentence referring to 4g was deleted.

36. The "Conclusions" section is mistakenly numbered "4" instead of "5"
**Response:** We change "4. Conclusions" to "5. Conclusions".

37. Figure 8: I think there is a typo in the caption, line 994; I believe "G6Solar" should be "G6Sulfur
**Response:** Thank you. We corrected it.

---

## Author Comment (AC2)

**Reviewer 2:**

The authors seek to quantify how geoengineering might be expected to affect wildfire occurrence, extent, and carbon emissions towards the end of the 21st century. They compare four different scenarios: two of the world without any geoengineering (one "cooler" - SSP2-4.5 - and one "warmer" - SSP5-8.5), one in which idealized (solar shade) geoengineering is applied to force global mean radiative forcing (RF) in the "warmer" scenario to follow that in the "cooler" scenario, and one in which stratospheric sulfates are emitted to achieve the same objective. To do so, they use the fully-coupled global chemistry-climate model WACCM, which includes an interactive treatment of wildfires. They evaluate wildfire outcomes principally through the metrics of total burned area and total carbon emissions, finding that - by both metrics - geoengineering "overcompensates" for the increase expected under SSP5-8.5, bringing total wildfires below the amount projected by SSP2-4.5. They also note some smaller differences in outcomes between the two geoengineering scenarios, and that there is some nonuniformity in the spatial distribution of outcomes.

The question asked by the authors is interesting, important, and timely, and the methods applied are appropriate. WACCM is a world-renowned climate model and the scenarios chosen are both well described and reasonably well known, making them more relevant to the community at large. The paper is incremental in nature, as it does not produce any astonishing new findings, but the content is nonetheless novel. The conclusions drawn are also supported by the data produced.

With this in mind, I have no major concerns regarding the content or framing of this paper. I have listed below some minor tehcnical and presentation concerns. Once these are addressed, I believe the manuscript will be appropriate for publication in ACP.

**Response:** Thank you for the comments and suggestions. Please see below for our response to the specific comments.

Minor technical issues

1. The authors appropriately caveat that the estimates of geoengineering's impacts are calculated based on small, variably-sized ensembles. In one of the central figures of the paper (Figure 6), correlations are presented between each "fire" variable (burned area and fire carbon emissions) and several "driver" variables (e.g. surface temperature). Values which are not significant at the p <= 0.1 level are not shown. However, it is not clear how significance is evaluated. Similarly, I am concerned by the differing numbers of simulations between all of the various scenarios. Both are important to this manuscript, and I recommend that the details be not only listed but also discussed in the manuscript main text.

**Response:** Thank you. All the calculations in the revised manuscript are now based on ensemble mean values regardless of the ensemble spread. Significance of the correlation coefficients is therefore calculated and evaluated based on the ensemble mean values. We added the following statement in Section 2.4 for clarification:

"To be consistent, for scenarios with multiple simulations, only ensemble means are shown and analyzed. I.e., ensemble means are calculated before any analyses or calculations, and hence a scenario with multiple simulations is treated in the same way as a scenario with only one simulation by only using the mean value of the ensemble members."

2. With regards to the question of the number of ensemble members used, I would recommend that the authors perform an analysis where a consistent ensemble size is used. Currently the manuscript appears to be biased by this discrepancy; for example, in Figure 2, the full range of burned areas between ensemble members is shown. However, smaller ensemble sizes will generally result in smaller ranges. Similarly, it would be useful to know how these ensemble members were initialized (e.g., in the geoengineering cases where there are 2 ensemble members, were they initialized with the same initial conditions as 2 of the 5 members from SSP5-8.5?).

**Response**: We agree with the reviewer that different ensemble sizes may result in biases in the ensemble ranges. I.e., a scenario with larger ensemble spread may be due to larger ensemble size rather than larger variabilities. Section 3 (Future trends of fires) shows ensemble spread, which is subject to this issue. In the contrast, in Section 4 (Mechanism of geoengineering impacting fires) we only use the ensemble mean values and the analyses does not involve ensemble spread therefore different ensemble sizes are unlikely to be an issue. To be consistent among scenarios and avoid differences in the range introduced by differences in the ensemble sizes, we changed the Section 3 and corresponding figures and tables (Tables 2 and 3) to only show ensemble mean values rather than spread. We added the following clarification in Section 2.4 where we describe the simulations:

"Different ensemble sizes could result in differences in ensemble spread. To be consistent, for scenarios with multiple simulations, only ensemble means are shown and analyzed. I.e., ensemble means are calculated before any analyses or calculations, and hence a scenario with multiple simulations is treat in the same way as a scenario with only one simulation by only using the mean value of the ensemble members."

We also added the following statement in Section 2.4 to describe the initialization of the ensemble members:

"The future projection simulations analyzed in this study were initialized with the ensemble WACCM6 historical simulations. Therefore, the initial conditions of different ensemble members are different."

3. On lines 209 to 216, the authors analyze how the estimated total burned area differs between SSPs. One specific claim stands out to me. The authors state that the changes under SSPs 8.5 (~20%) and 7.0 (~10%) are the largest, whereas those in the others scenarios are "relatively small". I would first request that qualitative claims such as this one be replaced with quantitative statements where possible, as I am unsure what "relatively small" means. However in this specific case I am also concerned that this conclusion may be driven more by the form of the evaluation than by a meaningful difference. Visually, it is unclear from Figure 2 that there is any significant difference between the trajectories taken by SSP1-2.6 and SSP3-7.0; however, these are also the two cases that had just one ensemble member. Providing simple quantitative statements (e.g. "increases of less than 4%" is better than "relatively small") would help, but better yet - as discussed above - would be a quantitative discussion of how confident the authors can be when comparing results from a single ensemble member to multi-member averages.

**Response**: Thank you. We have revised lines 209 to 216 to be more specific and quantitative. The revised Section 3.1 is as follows:

"The global total wildfire burned area in these simulations is projected to increase under all the SSP scenarios (Figure 1a). The largest increases (averages for the 2091-2100 period relative to the 2021-2030 period) in the global burned area are seen in the SSP5-8.5 scenarios (~20%). The

changes in SSP1-2.6 and SSP2-4.5 are less than 4% (see Table S2 for projected regional and global change of burned area and fire carbon emissions in 2091-2100 relative to 2021-2030 (%) under different scenarios). In terms of the spatial distribution, the 40°N–70°N latitude is the only latitude band in which the burned area consistently increases under all the SSP scenarios (Figure 1b). In the 10°S–5°N latitude band (tropical region), the burned area consistently decreases under all scenarios to a diverse extent. While global total burned area is expected to increase under most global warming scenarios, burned area may decrease in some regions due to changes in anthropogenic activities or reduced 2-m relative humidity and/or reduced soil moisture. A more detailed discussion on future trends of fire activity under the SSP scenarios are provided in the Supplement."

We also added the following discussion on the potential uncertainties to Section 2.4:

"Comparing results from a single simulation to multi-member averages could introduce potential uncertainties as ensemble mean values are in general different from values from a single member. However, the analyses and comparisons here are as useful as comparing single simulations, if not more, because in our approach we attempted to improve model projection for several scenarios by using ensemble means to replace single simulation values when possible."

**Minor presentation issues**

4. In several figures the color scales used are either confusing or misleading. The two most problematic examples are Figures 5 and 6. In Figure 5, the same color scale is used for 5 different quantities. Worse, for 4 of the quantities, "yellow" (i.e. the central value) is zero; however, for temperature, dark red is zero. This discrepancy is visually confusing. I would suggest using, at the very minimum, a different, one-sided color scale for the temperature change.

**Response:** Thank you for pointing this out. We have revised color scales in Figures 5 and 7 to make it clear. Please see below.

[Figure]

**Figure 5.** The difference in surface temperature (K) of (a) SSP2-4.5 from SSP5-8.5 (b) G6Solar from SSP5-8.5, (c) G6Sulfur from SSP5-8.5 averaged for 2091-2100. (d-f) are the same as (a-c) but for precipitation (mm/day). (g-i) are the same as (a-c) but for 2-meter relative humidity (%). (j-l) are the same as (a-c) but for 10-meter wind speed (m/s). (m-o) are the same as (a-c) but for soil water content at top 10 cm (kg/m$^2$). The grids where SSP2-4.5, G6Sulfur, or G6Solar is not significantly different from SSP5-8.5 is marked with white shade. Taking precipitation of SSP2-4.5 as an example, the significance for each model grid is calculated by student t-test (p value is 0.1) using 10 years of SSP2-4.5 precipitation data during 2091-2100 (10 data points) and 10 years of SSP5-8.5 precipitation data during 2091-2100 (10 data points).

[Figure]

**Figure 7**. The difference between G6Sulfur and G6Solar in (a) burned area fraction (BA; %/yr), (b) fire carbon emissions (Cemis; gC/m²/yr), (c) surface temperature (TS; K), (d) precipitation (Precip; mm/day), (e) 2-meter relative humidity (RH; %), (f) 10-meter wind speed (U10; m/s), (g) soil water content at top 10 cm (Soilwater; kg/m²), and (h) downwelling solar flux at the surface (FSDS; W/m²) averaged for 2091-2100. The grids where SSP2-4.5, G6Sulfur, or G6Solar is not significantly different from SSP5-8.5 is marked with white shade. Taking precipitation of SSP2-

4.5 as an example, the significance for each model grid is calculated by student t-test (p value is 0.1) using 10 years of SSP2-4.5 precipitation data during 2091-2100 (10 data points) and 10 years of SSP5-8.5 precipitation data during 2091-2100 (10 data points).

5. Similarly, the color scale used in Figure 6 is visually misleading. The extremes of the scale not only change from panel to panel, but the darkest shade changes from being a positive correlation of ~+0.5 (.e.g. 6a) to a negative correlation of -0.6 (6i). The color white also changes - while usually being 0, sometimes it is not (e.g. 6g, 6h). A consistent, two-sided color scale - for example that used in Figure 5 - running from -1 to +1 (or perhaps -0.6 to +0.6) would greatly help comprehension. Furthermore it is very difficult to read some of the values for the darkest backgrounds (see e.g. the EQAS/SSP2-4.5 value in 6i). Strangely the presentation in Figure S5 is much improved, although a standing issue with this manuscript is that a color scale with both red and green present is likely to cause issues for those with colorblindness.

**Response:** Thank you for pointing this out. We have revised color scales in Figures 6 (please see below) to make it consistent. And we have also revised all other Figures to make them colorblind friendly. The figures either use colorblind-friendly color scheme (e.g., Figures 1 and 2) or included information other than color for demonstration (e.g., numbers in Figure 6 and shapes in Figure 8).

[Figure]

**Figure 6.** Correlations of (a) surface temperature change (ΔTS) and burned area change for SSP2-4.5, G6Solar, and G6Sulfur, and (b) ΔTS and fire carbon emission change (ΔCemis) for SSP2-4.5, G6Solar, and G6Sulfur. Only correlations that are significant are labeled (p value <= 0.1). For SSP2-4.5, ΔTS is calculated for individual model grids within the region and annual values. It is defined as TS of SSP2-4.5 minus TS of SSP5-8.5 (the reference case). For G6Solar and G6Sulfur, ΔTS is defined in the same way as SSP2-4.5. ΔBA and ΔCemis are defined in the same way as ΔTS. (c-d) are the same as (a-b) but for precipitation change (ΔPrecip). (e-f) are the same as (a-b)

but for relative humidity change (ΔRH). (g-h) are the same as (a-b) but for 10-meter wind speed change (ΔU10). (i-j) are the same as (a-b) but for the change in soil water content at top 10 cm (ΔSOILWATER). Correlations are calculated for 14 fire regions (x-axis), following Giglio et al. (2010), namely Boreal North America (BONA), Temperate North America (TENA), Central America (CEAM), Northern Hemisphere South America (NHSA), Southern Hemisphere South America (SHSA), Europe (EURO), Middle East (MIDE), Northern Hemisphere Africa (NHAF), Southern Hemisphere Africa (SHAF), Boreal Asia (BOAS), Central Asia (CEAS), Southeast Asia (SEAS), Equatorial Asia (EQAS), and Australia and New Zealand (AUST). The definition of the regions can be found in Figure S3.

6. Why is the p-value of significance different between Figure 6 and Figure S5?

**Response**: Thank you for pointing this out. We updated Figure S5 so that its p value is consistent with Figure 6. Please see the updated figure below:

[Figure]

**Figure S5.** Correlations of burned area, fire carbon emissions with the driving factors (surface temperature, precipitation, relative humidity at 2 m, wind speed at 10 m, total vegetation carbon excluding carbon pool, and population) over 14 regions. The 14 regions are BONA (Boreal North America), TENA (Temperate North America), CEAM (Central America), NHSA (Northern Hemisphere South America), SHSA (Southern Hemisphere South America), EURO (Europe), MIDE (Middle East), NHAF (Northern Hemisphere Africa), SHAF (Southern Hemisphere Africa), BOAS (Boreal Asia), CEAS (Central Asia), SEAS (Southeast Asia), EQAS (Equatorial Asia), and AUST (Australia and New Zealand). The values of the correlations are labeled in the figure unless the correlation is not significant (P value > 0.1). The correlations are calculated based on the annual

mean values of the variables, and all simulations are included in the calculation regardless of their scenarios.

7. The caption for Figure 8 mentions that the shaded region is relevant to a comparison between delta-BA and delta-C emis, but that does not seem correct based on the axes. I suspect this was incorrectly copied from the caption for Figure 9?
**Response**: Thank you for pointing this out. There were typos in captions of both Figure 8 and 9. We have corrected them.

8. Lines 495-504: This phrasing is incorrect. The comparison being made is between strengths of correlation, not magnitudes of effects. Saying that "impacts of the shown variables (...) on burned area are in general stronger than their impacts on fire carbon emissions" implies that a conclusion is being drawn regarding the size of the impacts, not their degree of correlation with other variables. I would suggest rephrasing for clarity.
**Response**: We revised lines 495-504. Below is the revised paragraph:
"For G6Solar and G6Sulfur, the correlations of the shown variables (especially for $\Delta$TS, $\Delta$RH, $\Delta$U10, and $\Delta$FSDS) with burned area are in general stronger than their correlations with fire carbon emissions (as shown by more data points that fall into the shaded area). This is expected because these variables directly impact burned area, whereas fire carbon emissions are determined by both burned area and fuel availability. Fuel availability is further directly or indirectly impacted by many variables including but not limited to the shown ones here. Therefore, the correlations between the shown variables with fire carbon emissions are not as strong as their correlations with burned area. The patterns in G6Solar and G6Sulfur and closer to each other when using SSP2-4.5 as a reference (Figures 6). This is not only because their approaches to reducing forcing from SSP5-8.5 to 4.5 W/m$^2$ are different, but also because the scenario configuration of SSP2-4.5 is different from SSP5-8.5 and SSP5-8.5-based G6Solar and G6Sulfur (e.g., LULCC)."